# Capicua deficiency induces autoimmunity and promotes follicular helper T cell differentiation via derepression of ETV5

Sungjun Park[1,*], Seungwon Lee[2,*], Choong-Gu Lee[3,*], Guk Yeol Park[1], Hyebeen Hong[1], Jeon-Soo Lee[1], Young Min Kim[1], Sung Bae Lee[4], Daehee Hwang[5], Youn Soo Choi[6,7], John D. Fryer[8], Sin-Hyeog Im[1,2,3], Seung-Woo Lee[1,2] & Yoontae Lee[1,2]

High-affinity antibody production through the germinal centre (GC) response is a pivotal process in adaptive immunity. Abnormal development of follicular helper T ($T_{FH}$) cells can induce the GC response to self-antigens, subsequently leading to autoimmunity. Here we show the transcriptional repressor Capicua/CIC maintains peripheral immune tolerance by suppressing aberrant activation of adaptive immunity. CIC deficiency induces excessive development of $T_{FH}$ cells and GC responses in a T-cell-intrinsic manner. ETV5 expression is derepressed in *Cic* null $T_{FH}$ cells and knockdown of *Etv5* suppresses the enhanced $T_{FH}$ cell differentiation in *Cic*-deficient CD4$^+$ T cells, suggesting that *Etv5* is a critical CIC target gene in $T_{FH}$ cell differentiation. Furthermore, we identify *Maf* as a downstream target of the CIC–ETV5 axis in this process. These data demonstrate that CIC maintains T-cell homeostasis and negatively regulates $T_{FH}$ cell development and autoimmunity.

[1] Department of Life Sciences, Pohang University of Science and Technology, Pohang, Gyeongbuk 73673, Republic of Korea. [2] Division of Integrative Bioscience and Biotechnology, Pohang University of Science and Technology, Pohang, Gyeongbuk 73673, Republic of Korea. [3] Academy of Immunology and Microbiology, Institute for Basic Science, Pohang, Gyeongbuk 73673, Republic of Korea. [4] Department of Brain & Cognitive Sciences, Daegu Gyeongbuk Institute of Science and Technology (DGIST), Daegu 42988, Republic of Korea. [5] Center for Plant Aging Research, Institute for Basic Science, DGIST, Daegu 42988, Republic of Korea. [6] Department of Biomedical Sciences, Department of Medicine, Seoul National University College of Medicine, Seoul 03080, Republic of Korea. [7] Transplantation Research Institute, Department of Medicine, Seoul National University College of Medicine, Seoul 03080, Republic of Korea. [8] Department of Neuroscience, Mayo Clinic, Jacksonville, Florida 32224, USA. * These authors contributed equally to this work. Correspondence and requests for materials should be addressed to S.-H.I. (email: iimsh@postech.ac.kr) or to S.-W.L. (email: sw_lee@postech.ac.kr) or to Y.L. (email: yoontael@postech.ac.kr).

The germinal centre (GC) response is one of the most elegant processes in adaptive immunity and produce antibodies that have high affinity to antigens. In follicles, B cells that recognize antigens proliferate and form GCs. GCs are expanded by the proliferation of GC B cells and polarize into two microenvironments, the dark zone and the light zone[1–3]. GC B cells cycle between these two zones. In the dark zone, GC B cells rapidly proliferate and undergo somatic hypermutation, which enables antibody diversification and affinity maturation. In the light zone, GC B cells are selected on the basis of antigen affinity, undergo immunoglobulin class-switch recombination, and eventually give rise to antibody-secreting plasma cells or memory B cells[1–3]. During the GC response, several types of immune cell collaborate with B cells in the follicles, where follicular helper T ($T_{FH}$) cells have an instrumental function. $T_{FH}$ cells facilitate the selection and maturation of high-affinity GC B cells by multiple rounds of cognate interaction with B cells in the light zone; these interactions provide the selected B cells with crucial signals for survival and re-entry into the dark zone[3]. Because $T_{FH}$ cells have an important function in the generation of isotype-switched and affinity-matured antibodies, dysregulation of $T_{FH}$ cell development and function is closely associated with immunodeficiency-related pathogenesis or antibody-mediated autoimmune diseases including systemic lupus erythematosus[4–6].

Differentiation of $T_{FH}$ cells is initiated by the interaction of naïve T cells with dendritic cells (DCs), which, together with environmental factors, including cytokines, triggers expression of the chemokine receptor CXCR5 on DC-primed T cells[7,8]. The surface expression of CXCR5 enables T cells to migrate into B-cell follicles[7,8]. T cells that are targeted to enter B-cell follicles upregulate expression of the transcriptional repressor BCL6 and express an intermediate level of typical $T_{FH}$ molecules (for example, CXCR5, PD-1, ICOS and SAP) at the junction between T-cell and B-cell zone[9,10]. At this stage, developing $T_{FH}$ cells interact with cognate B cells and differentiate into GC $T_{FH}$ cells that express high levels of $T_{FH}$ molecules, such as PD-1 and CXCR5 (ref. 9). BCL6 as a master transcription factor for $T_{FH}$ cell differentiation[11–13] and BLIMP1 as an antagonist of BCL6 (ref. 11), plus several other transcription factors, help orchestrate $T_{FH}$ cell differentiation by exerting either a positive or negative effect, depending on the cellular context[8]. Among these factors, MAF (also known as c-MAF) was identified as a positive regulator of $T_{FH}$ cell differentiation in mice and humans. Maf deficiency decreases the frequency of CD4$^+$CXCR5$^+$ T cells in mice[14]. MAF regulates expression of Il21 in mouse $T_{FH}$ cells and, in conjunction with BCL6, MAF induces expression of CXCR4, CXCR5, PD-1, ICOS and IL-21 in human $T_{FH}$ cells[14–16]. Moreover, MAF expression is induced in CD4$^+$ T cells by ICOS co-stimulation[14] or by IL-6, an important cytokine for initiating $T_{FH}$ cell differentiation[15].

Capicua/CIC is a transcriptional repressor that is evolutionarily conserved from cnidarians to mammals[17], and it exists in short (CIC-S) and long (CIC-L) isoforms[17]. In mammals, CIC interacts with Ataxin-1/ATXN1 (ref. 18), of which polyglutamine (polyQ)-expanded form causes spinocerebellar ataxia type-1 (SCA1) neuropathogenesis, and its haploinsufficiency alleviates SCA1 progression[19]. Loss of the ATXN1–CIC complex results in hyperactivity, impaired learning and memory, and abnormal maturation and maintenance of upper-layer cortical neurons in mice[20]. CIC also suppresses the progression of several types of cancer[21–23]. CIC target genes that are critical for regulation of cancer progression include PEA3 group genes, ETV1/ER81, ETV4/PEA3 and ETV5/ERM, which are frequently overexpressed and promote tumourigenesis and metastasis in various types of cancer cell[21,24–26]. In vivo functions of CIC have been reported in studies of Cic hypomorphic (Cic-L$^{-/-}$) mice[19]. These mice die within 4 weeks after birth and have defects in lung alveolarization[27] and bile acid homeostasis[28]. However, CIC is expressed in most mouse tissues[27,28], so it may also have other important physiological functions. CIC levels are relatively high in the thymus[27,28] and in immune cells including T and NK cells (http://www.humanproteomemap.org)[29]. Moreover, our previous study shows that genes associated with immune activation and autoimmune diseases, such as SLE, asthma and thyroiditis, are significantly upregulated in the liver of Cic-L$^{-/-}$ mice[28]. These findings suggest that CIC may regulate immune responses and immune disorders.

Here we investigate CIC functions in immune system using various immune cell-specific Cic null mice. Our study finds previously unrecognised functions of CIC in regulation of T-cell activation and GC responses and suggests CIC as a key transcription factor in the suppression of autoimmunity.

## Results

**Autoimmunity in immune cell-specific Cic null mice.** To investigate in vivo requirement of CIC in homeostasis and functions of immune cells, we generated haematopoietic lineage cell-specific Cic null mice (Cic$^{f/f}$Vav1-Cre) by crossing mice carrying floxed Cic alleles (Cic$^{f/f}$) with Vav1-Cre mice (Supplementary Fig. 1a), and characterized them at 9–12 weeks of age. Development of lymphocytes in primary lymphoid organs was largely normal (Supplementary Fig. 1b,c), although an increase in proportion of thymic CD4$^+$FOXP3$^+$ regulatory T ($T_{reg}$) cells was observed (Supplementary Fig. 2).

Strikingly, all Cic$^{f/f}$Vav1-Cre mice developed enlarged secondary lymphoid organs, including spleens (Fig. 1a), mainly due to proliferation of B220$^+$ B cells (Fig. 1b). In addition, the number of CD8$^+$ T cells was significantly decreased, whereas that of macrophages and DCs was increased in spleen of Cic$^{f/f}$Vav1-Cre mice (Fig. 1b). The expression levels of several T-cell co-stimulatory ligands, such as CD80, CD86, CD40 and ICOSL, on the surface of DCs were comparable between Cic$^{f/f}$ (WT) and Cic$^{f/f}$Vav1-Cre mice (Supplementary Fig. 3). Cic$^{f/f}$Vav1-Cre mice had a higher frequency of effector/memory-phenotype (CD44$^{hi}$CD62L$^{lo}$) cells for both CD4$^+$ and CD8$^+$ subsets, compared with wild-type (WT) mice (Fig. 1c). Cic-deficient CD4$^+$CD44$^+$ T cells showed elevated expression of cell surface co-stimulatory molecules such as ICOS, PD-1, OX40 and GITR (Supplementary Fig. 4). Flow cytometry analysis on cytokine expression profiles revealed that the frequency of IFN$\gamma^+$, TNF$^+$, IL-21$^+$, IL-13$^+$ and IL-22$^+$ CD4$^+$ T cells was significantly increased in spleen of Cic$^{f/f}$Vav1-Cre mice (Supplementary Fig. 5a). Cic$^{f/f}$Vav1-Cre mice also had a higher frequency of CCR4$^-$CXCR3$^+$ ($T_H$1-related) and CCR4$^+$CCR6$^-$ ($T_H$2-related) CD4$^+$ T cells[30] (Supplementary Fig. 5b). Thus, loss of CIC spontaneously activates T cells, resulting in accumulation of effector/memory cells with mixed T helper ($T_H$) phenotypes in mice. On the other hand, Cic null naïve CD4$^+$ T cells differentiated into each $T_H$ subset with similar efficiency to WT cells under in vitro polarizing conditions (Supplementary Fig. 6), suggesting that CIC might not be a crucial factor for each $T_H$ subset differentiation.

Cic$^{f/f}$Vav1-Cre mice had a higher proportion of CD4$^+$CD25$^-$FOXP3$^+$ T cells in spleen, but a comparable proportion of CD4$^+$CD25$^+$FOXP3$^+$ T cells, compared with WT mice (Supplementary Fig. 7a). Expression levels of surface molecules that are critical for controlling $T_{reg}$ cell homeostasis and function, such as GITR, CTLA-4, CD103 and GARP (refs 31,32), were comparable between WT and Cic null CD4$^+$FOXP3$^+$ T cells regardless of CD25 expression (Supplementary Fig. 7b). Consistent with this result, WT and Cic-deficient CD4$^+$CD25$^+$

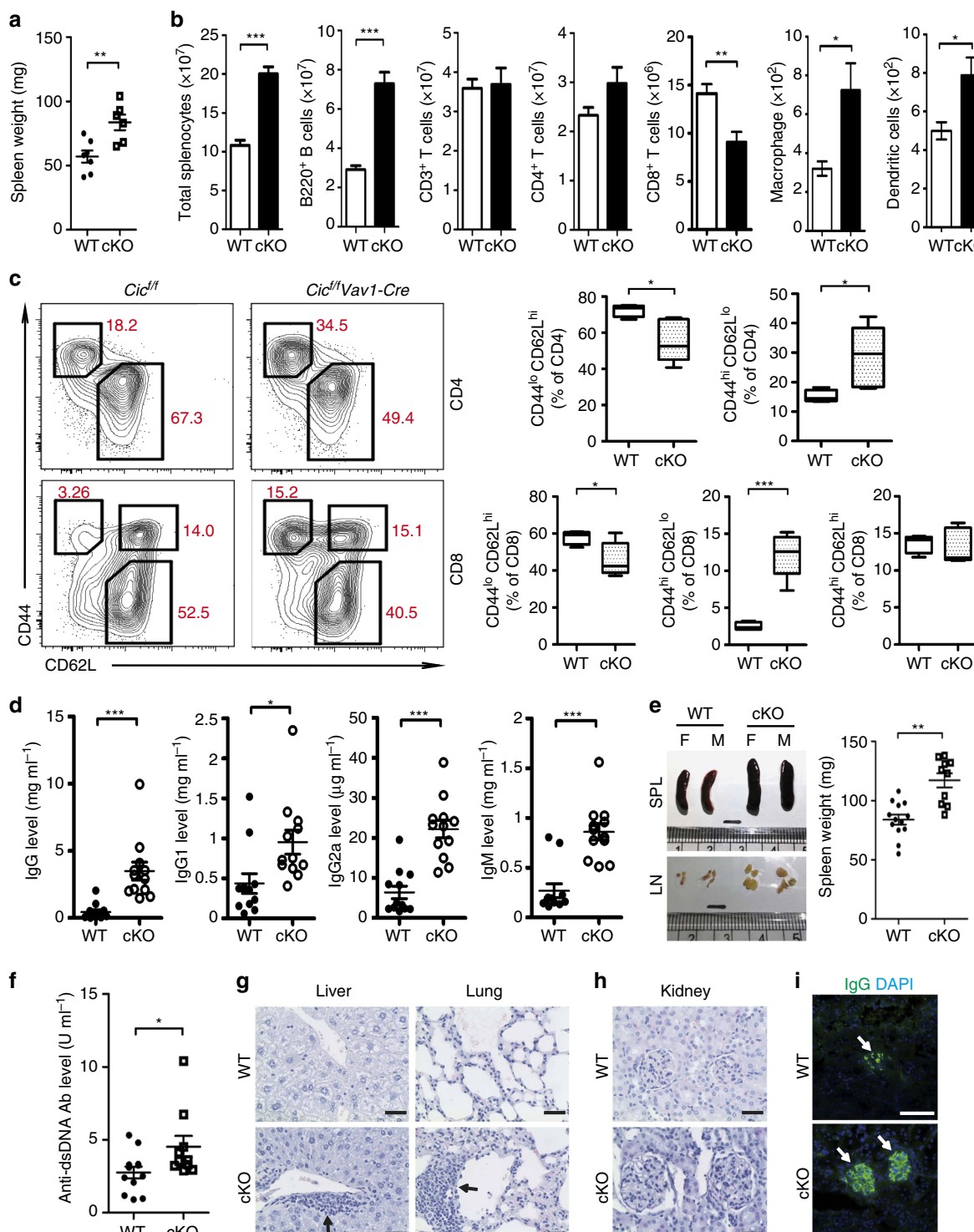

**Figure 1 | Lymphoproliferative autoimmunity in immune cell-specific *Cic* null mice.** (**a**,**b**) Spleen weights (**a**) and the numbers of total splenocytes, B220+ B, CD3+ T, CD4+ T, CD8+ T, macrophages (CD11b+CD11c−F4/80+) and dendritic cells (CD11b+CD11c+MHCII+) (**b**) in 10-week-old *Cic*f/f (WT) and *Cic*f/f*Vav1-Cre* (cKO) mice. (**c**) FACS analysis showing the proportion of CD44loCD62Lhi naïve T cells and CD44hiCD62Llo effector/memory T cells in spleen of *Cic*f/f and *Cic*f/f*Vav1-Cre* mice. All data are representative of two independent experiments. Data are presented as box-and-whisker plots. *n* = 4–5 per each genotype. (**d**) Serum levels of total IgG, IgG1, IgG2a and IgM in 12-week-old *Cic*f/f and *Cic*f/f*Vav1-Cre* mice were measured by ELISA. (**e**) Images of dissected spleens (SPL) and draining lymph nodes (LN) from *Cic*f/f and *Cic*f/f*Vav1-Cre* mice at the age of 12 months. The graph shows average weights of spleens from the *Cic*f/f and *Cic*f/f*Vav1-Cre* mice. M, male; F, female. (**f**) Serum levels of anti-dsDNA antibody in the 12-month-old *Cic*f/f and *Cic*f/f*Vav1-Cre* mice were measured by ELISA. The graphs in **a**,**b**,**d**–**f** show data as mean ± s.e.m. and each dot in graphs represents an individual mouse. *P < 0.05, **P < 0.01 and ***P < 0.001 (two-tailed two-sample unequal variance Student *t*-test). (**g**) Immune cell infiltration in liver and lung from the 12-month-old *Cic*f/f*Vav1-Cre* mice. Tissue sections were stained with haematoxylin and eosin (H&E). Arrows indicate immune cell infiltrates. (**h**) H&E-stained kidney sections showing glomerulonephritis in the 12-month-old *Cic*f/f*Vav1-Cre* mice. (**i**) Immunostaining for IgG deposition in kidney glomeruli of the 12-month-old *Cic*f/f*Vav1-Cre* mice. Representative images from two independent experiments are shown. (**g**–**i**) Scale bars, 100 μm.

$T_{reg}$ cells had a comparable suppressive activity on IL-2 secretion and proliferation of responding $CD4^+CD25^-$ T cells (Supplementary Fig. 7c,d). The numbers of $CD8^+CD25^+$ $FOXP3^+$ and $CD8^+ICOSL^+CXCR5^+$ (Qa-1-restricted $CD8^+$ $T_{reg}$ cells)[33,34] T cells were also comparable between WT and $Cic^{f/f}Vav1\text{-}Cre$ mice (Supplementary Fig. 7e,f). These results, together with the increased frequency of thymic $T_{reg}$ cells (Supplementary Fig. 2), indicate that the T cell hyperactivation in $Cic^{f/f}Vav1\text{-}Cre$ mice was not due to defects in $T_{reg}$ cell compartment.

Because the abnormal activation of T cells and the increased frequency of $CD4^+CD25^-FOXP3^+$ T cells are often associated with autoimmunity[35,36], we assessed whether $Cic^{f/f}Vav1\text{-}Cre$ mice display autoimmune phenotypes with age. The $Cic^{f/f}Vav1\text{-}Cre$ mice developed hyperglobulinemia at 12 weeks of age (Fig. 1d), and exhibited several typical autoimmune attributes at around 1 year of age, including enlarged secondary lymphoid organs (Fig. 1e), increased serum levels of anti-dsDNA antibody (Fig. 1f), infiltration of immune cells into tissues (Fig. 1g), glomerulonephritis (Fig. 1h) and IgG deposition in kidney glomeruli (Fig. 1i). Together, these findings demonstrate that CIC is indispensable for suppression of lymphoproliferative autoimmunity.

**Spontaneous induction of GC responses by CIC deficiency.** Increased frequencies of $T_{FH}$-like cells are often observed in patients with autoimmune diseases including SLE[6], and studies of $Roquin^{san}$ mice show that overexpression of IFNγ and ICOS, a co-stimulatory molecule highly expressed in $T_{FH}$ cells[37,38], promotes $T_{FH}$ cell development, which in turn spontaneously induces a lupus-like autoimmune syndrome[39,40]. It is also known that IL-21 and IL-4 are key cytokines that $T_{FH}$ cells express during the GC response[41]. Because $Cic^{f/f}Vav1\text{-}Cre$ mice had the increased T-cell surface expression of ICOS, the increased proportions of $CD4^+IFNγ^+$, $CD4^+IL\text{-}21^+$ and $CD4^+IL\text{-}4^+$ T cells, and the autoimmune-like symptoms similar to those in $Roquin^{san}$ mice[42], we set out to determine whether the $T_{FH}$ subset was elevated in $Cic^{f/f}Vav1\text{-}Cre$ mice. The frequencies of $T_{FH}$ ($PD\text{-}1^{int}CXCR5^{int}$) and GC $T_{FH}$ ($PD\text{-}1^{hi}CXCR5^{hi}$) cells were significantly increased in the spleen of 9-week-old $Cic^{f/f}Vav1\text{-}Cre$ mice at the expense of non-$T_{FH}$ cells (Fig. 2a). The increased $T_{FH}$ cell frequency in $Cic^{f/f}Vav1\text{-}Cre$ mice was confirmed by flow cytometry analysis of $CD4^+BCL6^+CXCR5^+$ T cells (Fig. 2b). Consistent with these results, the proportion of $CD4^+ICOS^+$ T cells was also markedly increased (Fig. 2c). On the other hand, levels of BCL6 in $PD\text{-}1^+CXCR5^+$ $T_{FH}$ cells were comparable between WT and $Cic^{f/f}Vav1\text{-}Cre$ mice (Fig. 2d). The frequency of $IFNγ^+$ $T_{FH}$ cells was substantially increased in $Cic^{f/f}Vav1\text{-}Cre$ mice, whereas that of $IL\text{-}21^+$ and $IL\text{-}4^+$ $T_{FH}$ cells was comparable between WT and $Cic^{f/f}Vav1\text{-}Cre$ mice (Fig. 2e). Because the primary function of $T_{FH}$ cells is to help B cells form GC reactions, we further analysed GC B ($B220^+Fas^+GL\text{-}7^+$) cells. The proportion of GC B cells was significantly higher in $Cic^{f/f}Vav1\text{-}Cre$ mice than in WT mice (Fig. 2f). Taken together, these data indicate that CIC is required for suppression of spontaneous induction of GC reactions.

**T-cell-intrinsic functions of CIC.** Because several abnormalities in T cells were observed in $Cic^{f/f}Vav1\text{-}Cre$ mice, we next examined CIC in T cells. CIC protein levels gradually increased along with the activation of $CD4^+$ T cells by anti-CD3 and anti-CD28 antibodies (Fig. 3a), and were higher in effector/memory $CD4^+$ T cells than in naïve $CD4^+$ T cells (Fig. 3b). $Cic$-deficient $CD4^+$ T cells secreted IL-2 and proliferated more efficiently than WT T cells when stimulated with anti-CD3 (Fig. 3c,d), suggesting

that CIC negatively regulates an activation signal that is mediated by T-cell receptors (TCRs). However, this difference was largely abrogated by addition of anti-CD28 (Fig. 3c,d), indicating that a strong co-stimulatory signal through CD28 overrides the enhanced TCR response in $Cic$-deficient T cells.

To determine the *in vivo* consequences of CIC deficiency in T cells, we generated mice with $Cic$ deletion specific to T cells ($Cic^{f/f}Cd4\text{-}Cre$) (Fig. 4a). Development of T cells in thymus was largely normal in $Cic^{f/f}Cd4\text{-}Cre$ mice (Supplementary Fig. 8a). Given that CIC deficiency induced autoimmunity and that $Cic$ null $CD4^+$ T cells more robustly responded to TCR stimulation than WT cells *in vitro* (Fig. 3c,d), we assessed thymic negative selection, which is a critical process that removes autoreactive T cells in thymus[43], in WT and $Cic^{f/f}Cd4\text{-}Cre$ mice. TCR-induced apoptosis comparably occurred in WT and $Cic$ null double positive (DP) thymocytes (Supplementary Fig. 8b), suggesting that TCR-induced negative selection might not be affected by CIC deficiency. Consistent with this finding, WT and $Cic$ null $CD4^+$ thymocytes exhibited a comparable induction of Nur77, which has been implicated in negative selection through its ability to convert BCL2 into a proapoptotic molecule[44,45], in response to TCR stimulation (Supplementary Fig. 8c).

On the other hand, as observed in the $Cic^{f/f}Vav1\text{-}Cre$ mice, hyperglobulinemia, T-cell hyperactivation and increased populations of $CD25^-FOXP3^{+-}$, $T_H1$-type and $T_H2$-type $CD4^+$ T cells in spleen occurred in $Cic^{f/f}Cd4\text{-}Cre$ mice at 12 weeks of age (Fig. 4b, Supplementary Fig. 9). These mice also showed several systemic autoimmune phenotypes with age, including increased serum levels of anti-dsDNA antibody (Fig. 4c), infiltration of immune cells into tissues (Fig. 4d) and glomerulonephritis (Fig. 4e). Moreover, the proportions of $T_{FH}$, GC $T_{FH}$, $CD4^+ICOS^+$ T and GC B cells were significantly higher in $Cic^{f/f}Cd4\text{-}Cre$ mice than in WT mice (Fig. 4f–h). The frequency of IFNγ-expressing $T_{FH}$ cells was increased in the spleen of $Cic^{f/f}Cd4\text{-}Cre$ mice (Supplementary Fig. 10a). Because IFNγ drives IgG2a class-switching in B cells during GC reactions[40,46,47], we further assessed $IgG2a^+$ GC B cells. As expected, the frequency and the number of $IgG2a^+$ GC B cells were also increased in $Cic^{f/f}Cd4\text{-}Cre$ mice (Supplementary Fig. 10b). Taken together, these findings demonstrate that the spontaneous inductions of T-cell activation, GC response and systemic autoimmunity were due to loss of CIC in T cells.

Follicular regulatory T ($T_{FR}$) cells, defined as $CD4^+$ $FOXP3^+$ $PD\text{-}1^+CXCR5^+$ cells, are a specialized subset of regulatory T cells that inhibit antibody production[48–50]. Because the proportion of $CD4^+FOXP3^+$ T cells increased in $Cic$-deficient mice (Supplementary Figs 7a and 9b), we determined whether $T_{FR}$ cells were increased in $Cic^{f/f}Cd4\text{-}Cre$ mice. As with $CD4^+$ $FOXP3^-PD\text{-}1^+CXCR5^+$ $T_{FH}$ cells, $CD4^+FOXP3^+PD\text{-}1^+$ $CXCR5^+$ $T_{FR}$ cells were also significantly overrepresented in the spleen of $Cic^{f/f}Cd4\text{-}Cre$ mice compared with WT mice (Fig. 4i). Importantly, the ratio $T_{FH}/T_{FR}$, which is a critical factor that dictates the magnitude of antibody production[50], was also increased in the $Cic^{f/f}Cd4\text{-}Cre$ mice (Fig. 4j), consistent with the observation that GC B cells and serum immunoglobulin levels were elevated in $Cic^{f/f}Cd4\text{-}Cre$ mice (Fig. 4b,h).

**Normal T-cell homeostasis in $T_{reg}$-specific $Cic$ null mice.** Although the suppressive activity toward conventional T cells and expression profiles of $T_{reg}$ cell-associated surface molecules were comparable between WT and $Cic$-deficient $CD4^+$ $T_{reg}$ cells (Supplementary Fig. 7b–d), the frequency of $CD4^+CD25^-$ $FOXP3^+$ T cells was substantially increased in $Cic$-deficient mice (Supplementary Figs 7a and 9b). To directly address the function of CIC in development and function of $FOXP3^+$ $T_{reg}$ cells,

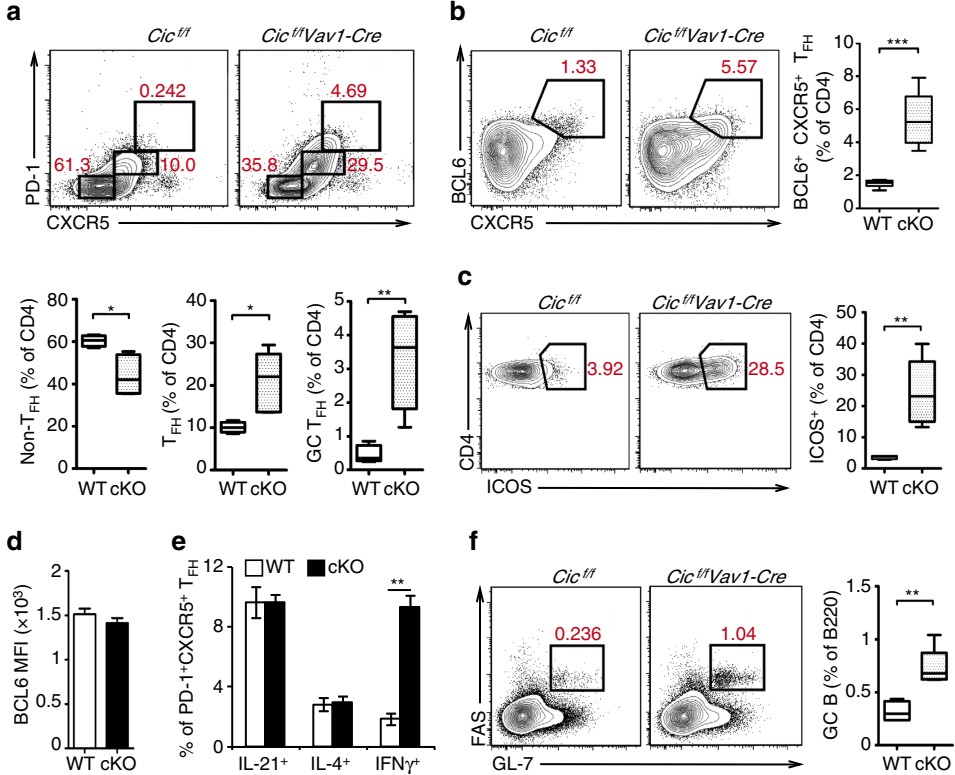

**Figure 2 | Increased proportion of T_FH and GC B cells in *Cic^f/f Vav1-Cre* mice.** (**a–c,f**) Flow cytometry analyses of non-T_FH, T_FH and GC T_FH cells (**a**), CD4^+BCL6^+CXCR5^+ T_FH cells (**b**), CD4^+ICOS^+ T cells (**c**) and GC B cells (**f**) in spleen from 9-week-old *Cic^f/f* (WT) and *Cic^f/f Vav1-Cre* (cKO) mice. All data are representative of three independent experiments with $n = 4$–5 mice per group in each experiment. Numbers adjacent to outlined areas indicate per cent of each cell population among splenic CD4^+ or B220^+ cells. Data are presented as box-and-whisker plots. *$P < 0.05$, **$P < 0.01$ and ***$P < 0.001$ (two-tailed two-sample unequal variance student *t*-test). (**d**) Mean fluorescence intensity (MFI) of BCL6 expression in CD4^+PD-1^+CXCR5^+ T_FH cells from 9-week-old *Cic^f/f* and *Cic^f/f Vav1-Cre* mice. $n = 3$ mice per each genotype. (**e**) Flow cytometry analysis of IL-21, IL-4 or IFNγ-expressing T_FH cells in spleen of 9-week-old *Cic^f/f* and *Cic^f/f Vav1-Cre* mice. The proportions of each cytokine-expressing T_FH cells are presented. (**d,e**) $n = 3$ mice per each genotype. The bar graphs show data as mean ± s.e.m. **$P < 0.01$ (two-tailed two-sample unequal variance Student *t*-test).

we generated FOXP3^+ T_reg cell-specific *Cic* null (*Cic^f/f Foxp3-YFP-Cre*) mice by crossing *Cic^f/f* mice with *Foxp3-YFP-Cre* mice[51]. The ablation of *Cic* alleles in *Foxp3*-expressing (YFP^+) cells did not affect formation of CD25^+FOXP3^+ and CD25^−FOXP3^+ CD4^+ T cells in both thymus and spleen (Supplementary Fig. 11a–c). The proportion of naïve and effector/memory cells for both CD4^+ and CD8^+ subsets, T_FH, T_FR and GC B cells was also comparable between *Cic^+/+ Foxp3-YFP-Cre* (WT) and *Cic^f/f Foxp3-YFP-Cre* mice (Supplementary Fig. 11d–f). These results suggest that CIC deficiency in FOXP3^+ T_reg cells does not cause defects in development and function of FOXP3^+ T_reg cells.

Since the FOXP3^+ T_reg cell-specific *Cic* deletion did not promote formation of CD4^+CD25^−FOXP3^+ T-cell population in spleen (Supplementary Fig. 11c), we examined expression of FOXP3 in conventional CD4^+ T cells from WT and *Cic^f/f Cd4-Cre* mice. Interestingly, the proportion of FOXP3-expressing naïve (CD45RB^hiCD44^loCD62L^hi) CD4^+ T cells was markedly increased in the spleen of *Cic^f/f Cd4-Cre* mice, compared with WT mice (Supplementary Fig. 12a). Consistent with this result, CD4^+CD25^−FOXP3^+ T cells in *Cic^f/f Cd4-Cre* mice were composed of both CD44^loCD62L^hi and CD44^hiCD62L^lo populations, while those in WT mice were mainly CD44^hiCD62L^lo cells (Supplementary Fig. 12b). We also determined the origin of CD4^+CD25^−FOXP3^+ T cells in *Cic^f/f Cd4-Cre* mice by flow cytometry analysis for expression of Helios and Neurophilin-1/NRP1, both of which are markers for thymus-derived T_reg (tT_reg) cells[52–54]. The frequency of Helios^−NRP1^− cells for both

CD25^+FOXP3^+ and CD25^−FOXP3^+ T-cell subsets was significantly increased in the spleen of *Cic^f/f Cd4-Cre* mice (Supplementary Fig. 12c), demonstrating that the increased frequency of CD4^+CD25^−FOXP3^+ T cells in *Cic^f/f Cd4-Cre* mice was due to the expansion of peripherally-induced T_reg (pT_reg) cells. Taken together, our data suggest that CIC deficiency induces FOXP3 expression in naïve CD4^+ T cells, thereby increasing the population of CD4^+CD25^−FOXP3^+ T cells in the periphery.

**Regulation of T_FH cell differentiation by the CIC–ETV5 axis.** To understand the molecular mechanism mediating the CIC deficiency-induced increase in T_FH cells, we set out to identify CIC target genes in CD4^+ T cells. We analysed gene expression profiles in naïve and anti-CD3/CD28 antibody-activated CD4^+ T cells from WT and *Cic^f/f Cd4-Cre* mice by RNA sequencing. Among the genes significantly upregulated (log_2(fold change) > 1.5; $P < 0.05$) in *Cic*-deficient CD4^+ T cells, only *Etv4* and *Etv5* were upregulated irrespective of the status of T-cell activation (Supplementary Data 1). Both *Etv4* and *Etv5* have been well-characterized as CIC target genes in various tissues[19,22,24,27]. qRT-PCR analysis for mRNA levels of Pea3 group genes (*Etv1, Etv4* and *Etv5*) (Fig. 5a) and quantitative PCR (qPCR) analysis for Pea3 group gene promoter regions in total CIC-associated DNA fragments (Fig. 5b) verified that *Etv4* and *Etv5* are direct targets of CIC in CD4^+ T cells.

We then examined the levels of *Etv4* and *Etv5* in CD4^+ PD-1^+CXCR5^+ T_FH cells from WT and *Cic^f/f Cd4-Cre* mice by

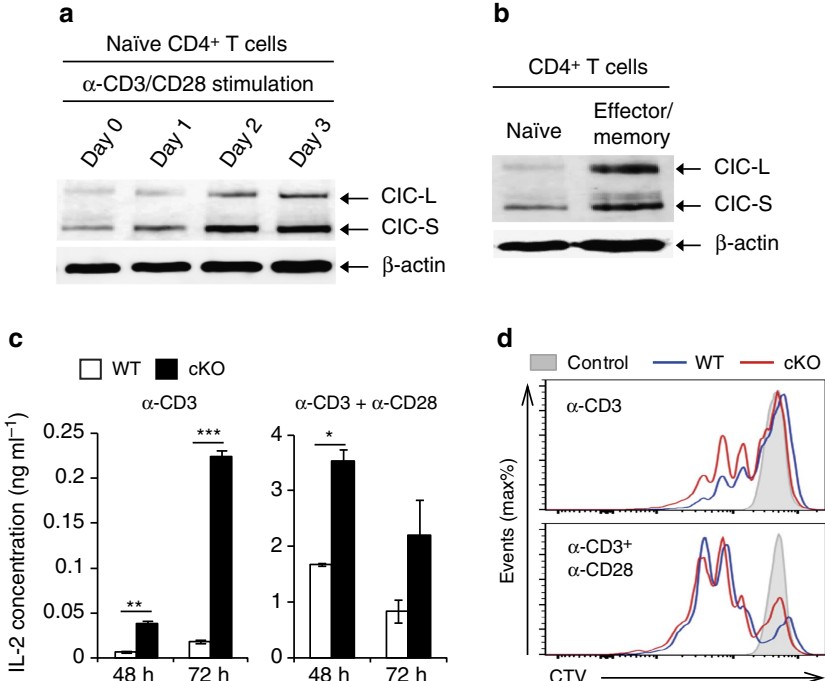

**Figure 3 | CIC negatively regulates T-cell activation.** (**a,b**) Increased CIC expression in activated CD4⁺ T cells. Western blot analysis for CIC levels in CD4⁺ T cells during T-cell activation by α-CD3/CD28 antibodies (**a**) and in sorted naïve (CD44$^{lo}$CD62L$^{hi}$) and effector/memory (CD44$^{hi}$CD62L$^{lo}$) CD4⁺ T cells (**b**). (**c**) ELISA of IL-2. WT and *Cic*-deficient (cKO) CD4⁺CD25⁻CD44$^{lo}$CD62L$^{hi}$ naïve T cells were stimulated with plate-bound anti-CD3 (1.0 μg ml⁻¹) in the presence (right) or absence (left) of plate-bound anti-CD28 (2.0 μg ml⁻¹). The supernatants were taken 48 h and 72 h after stimulation and subjected to ELISA for IL-2 concentration. $n = 4$ per each sample. Error bars indicate s.e.m. *$P < 0.05$, **$P < 0.01$ and ***$P < 0.001$ (two-tailed two-sample unequal variance Student *t*-test). (**d**) In vitro T-cell proliferation assay. Naïve CD4⁺ T cells purified from pooled spleens and lymph nodes of *Cic^{f/f}* and *Cic^{f/f}Vav1-Cre* mice were labelled with CTV dye and stimulated with plate-bound anti-CD3 (1.0 μg ml⁻¹) in the presence (right) or absence (left) of plate-bound anti-CD28 (2.0 μg ml⁻¹). The cells were analysed 72 h after stimulation. Data are representative of three independent experiments. Shaded area: unstimulated control, blue line: *Cic^{f/f}*, red line: *Cic^{f/f}Vav1-Cre*.

qRT-PCR analysis. Consistent with the previous result (Fig. 5a), levels of both genes were markedly increased in *Cic*-deficient T$_{FH}$ cells (Fig. 5c), indicating that their expression was derepressed in T$_{FH}$ cells in the absence of CIC. Interestingly, however, only ETV5 expression, but not ETV4 expression, was substantially increased in *Cic*-deficient T$_{FH}$ cells at the protein level (Fig. 5d), suggesting that ETV4 expression is tightly controlled in T$_{FH}$ cells at the posttranscriptional level. We also found that ETV5 levels were higher in T$_{FH}$ cells than in non-T$_{FH}$ cells (Fig. 5c,e), whereas CIC levels in the two T-cell populations were comparable (Fig. 5e), implying that ETV5 might be critical for differentiation or function, or both, of T$_{FH}$ cells. To further explore this possibility, we performed adoptive transfer experiments using OT-II cells, which express ovalbumin (OVA)-specific TCRs[55]. Thy1.1⁺OT-II cells were infected with control or ETV5-expressing retrovirus (Supplementary Fig. 13a) and adoptively transferred to Thy1.2⁺ recipient mice. Immunization of the host mice with 4-hydroxyl-3-nitrophenyl (NP)-OVA in alum resulted in generation of a higher percentage of T$_{FH}$ cells (Fig. 5f), demonstrating that ETV5 promotes T$_{FH}$ cell differentiation. Consistent with this result, knockdown of ETV5 in OT-II cells by shRNA against *Etv5* (shETV5) suppressed T$_{FH}$ cell differentiation (Fig. 5g). Lastly, we conducted the adoptive transfer experiment to examine whether the increased T$_{FH}$ cells in *Cic* mutant mice was due to derepression of ETV5. WT and *Cic*-deficient OT-II cells infected with control or shETV5-expressing retrovirus were adoptively transferred to Thy1.2⁺ recipient mice. Seven days after immunization with NP-OVA in alum, the donor OT-II cells were analysed for T$_{FH}$ cell differentiation by flow cytometry. As expected from previous results (Figs 2a,b and 4f), T$_{FH}$ cell

differentiation was more prominent with *Cic*-deficient donor OT-II cells than with WT OT-II cells (Figs 5h and 6f). Strikingly, such enhanced T$_{FH}$ cell differentiation of *Cic*-deficient OT-II cells was blunted when ETV5 expression was knocked-down with shETV5 (Fig. 5h, Supplementary Fig. 13b). Altogether, these data demonstrate that the CIC–ETV5 axis is crucial for regulation of T$_{FH}$ cell differentiation.

**Maf as a critical ETV5 target in T$_{FH}$ cell differentiation.** ETV5 function in T$_{FH}$ cell development has not been investigated. To comprehensively understand how the CIC–ETV5 axis regulates differentiation of T$_{FH}$ cells, we sought to identify ETV5 target genes that could be involved in this process. We initially analysed expression profiles of several T$_{FH}$-related genes in non-T$_{FH}$ and T$_{FH}$ cells from WT and *Cic^{f/f}Cd4-Cre* mice. *Bcl6, Batf, Maf, Cxcr5* and *Ifng* were significantly increased, whereas *Lef1* and *Foxp1* were decreased in WT T$_{FH}$ cells compared with WT non-T$_{FH}$ cells (Fig. 6a). Importantly, *Batf, Maf, Icos* and *Ifng* were significantly upregulated in *Cic*-deficient T$_{FH}$ cells compared with WT T$_{FH}$ cells; the increase was greatest in *Maf*, which promotes T$_{FH}$ cell differentiation and GC reactions[14,16] (Fig. 6a). We confirmed the overexpression of MAF in *Cic*-deficient T$_{FH}$ cells at the protein level (Fig. 6b).

We also examined which T$_{FH}$-related genes are regulated by ETV5 in CD4⁺ T cells. Anti-CD3/CD28 antibody-activated CD4⁺ T cells were infected with control or ETV5-expressing retrovirus and cultured in the presence of IL-6 and IL-21 (T$_{FH}$-like condition)[56]. Among the genes tested, only *Maf* levels were substantially increased by ETV5 overexpression (Fig. 6c).

Consistent with this, it is known that *Maf* is a direct target gene of ETV5 in ocular lens cells[57]. We confirmed that ETV5 directly bound to *Maf* promoter in IL-6-treated CD4$^+$ T cells by chromatin immunoprecipitation (ChIP)-qPCR analysis (Supplementary Fig. 14). In contrast, in the absence of IL-6 and

IL-21, ETV5 overexpression did not significantly increase levels of *Maf* (Fig. 6d), suggesting that ETV5 might induce expression of *Maf* during T$_{FH}$ cell development. These results were also confirmed by western blot analysis (Fig. 6e). Because IL-6 and IL-21 can activate STAT3 pathway, we examined whether

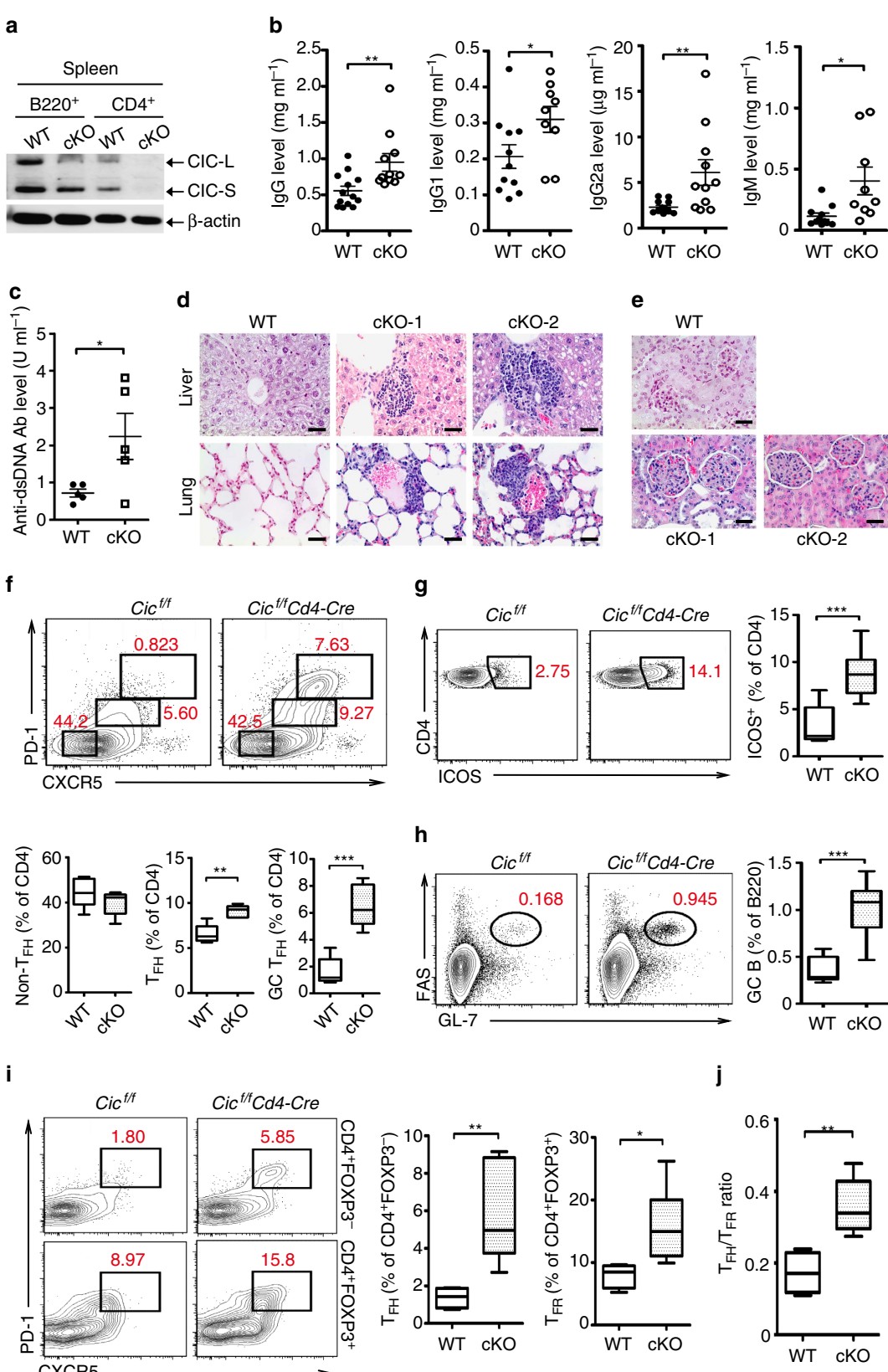

activation of STAT3 is required for ETV5-mediated induction of *Maf* expression. Treatment with AG490 (inhibitor of JAK2-STAT3 pathway)[58] and Stattic (inhibitor for STAT3

dimerization)[59] completely suppressed the induction of *Maf* expression by ETV5 in CD4$^+$ T cells incubated under T$_{FH}$-like condition (Supplementary Fig. 15a,b). However, treatment

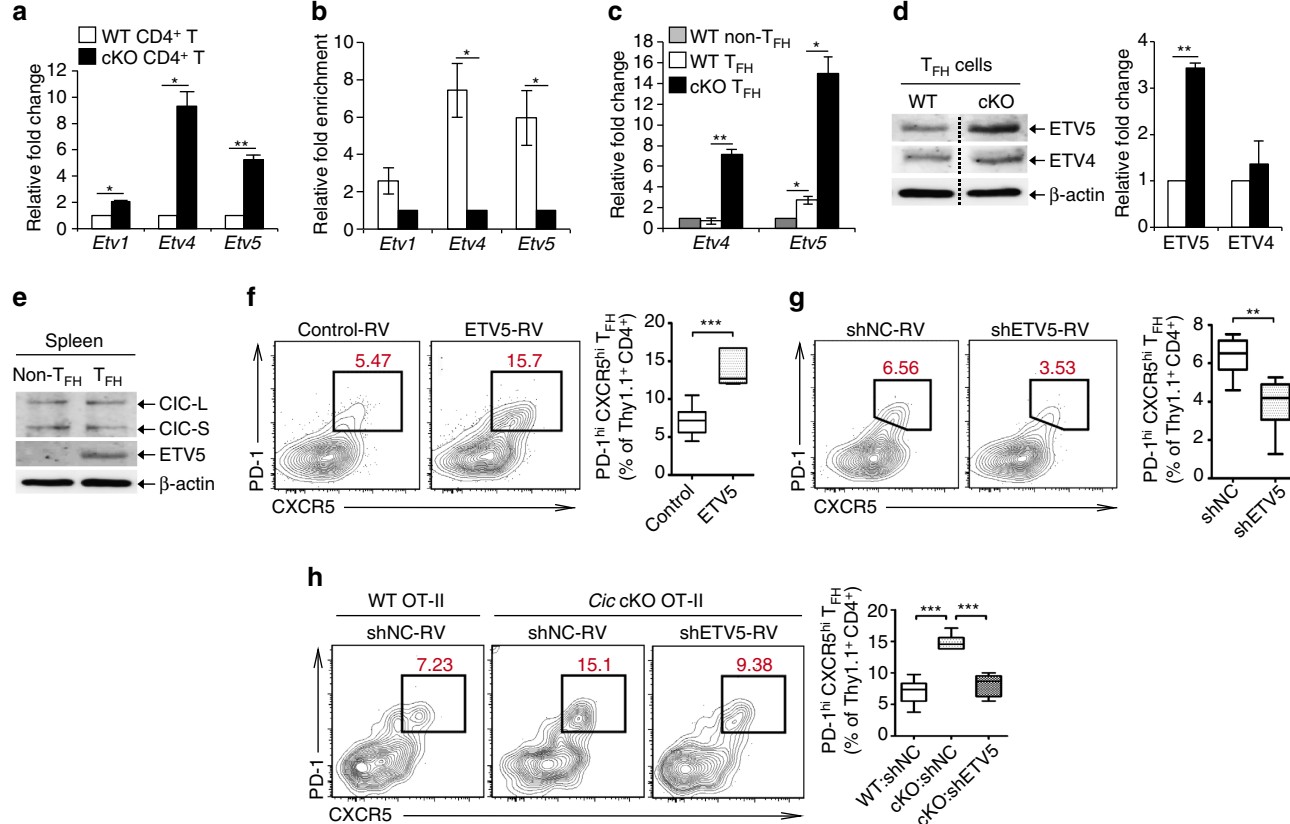

**Figure 5 | De-repression of *Etv5* promotes T$_{FH}$ cell differentiation in *Cic*-deficient CD4$^+$ T cells.** (**a**) qRT-PCR analysis for levels of *Pea3* group genes in splenic CD4$^+$ T cells from *Cic*$^{f/f}$ (WT) and *Cic*$^{f/f}$*Cd4-Cre* (cKO) mice. $n=3$ per each genotype. (**b**) Chromatin immunoprecipitation (ChIP)-qPCR analysis showing CIC promoter occupancy of *Etv4* and *Etv5* in CD4$^+$ T cells. CD4$^+$ T cells purified from spleen of *Cic*$^{f/f}$ and *Cic*$^{f/f}$*Cd4-Cre* mice were used. $n=3$ per each genotype. (**c**) qRT-PCR analysis for levels of *Etv4* and *Etv5* in sorted CD4$^+$PD-1$^-$CXCR5$^-$ non-T$_{FH}$ and CD4$^+$PD-1$^+$CXCR5$^+$ T$_{FH}$ cells. Splenocytes from 5 to 6 *Cic*$^{f/f}$ mice or 3 to 4 *Cic*$^{f/f}$*Cd4-Cre* mice were pooled and subjected to cell sorting. Three independent experiments were conducted. (**d**) Western blot analysis for levels of ETV4 and ETV5 in sorted splenic CD4$^+$PD-1$^+$CXCR5$^+$ T$_{FH}$ cells. The dash lines on blot images indicate cropping without image manipulation to either side. Three independent experiments were conducted. The graphs in **a–d** show data as mean ± s.e.m. *$P<0.05$ and **$P<0.01$ (two-tailed two-sample unequal variance Student *t*-test). (**e**) Western blot analysis for levels of CIC and ETV5 in sorted non-T$_{FH}$ and T$_{FH}$ cells from spleen of WT C57BL/6 (B6) mice. The images are representative of two independent experiments. (**f**) Thy1.1$^+$ OT-II cells infected with control or ETV5-expressing retrovirus were adoptively transferred to Thy1.2$^+$ B6 recipient mice. Eight days after immunization with NP-OVA in alum, donor cells were analysed for T$_{FH}$ cell differentiation using flow cytometry. Control-RV, $n=12$; ETV5-RV, $n=9$. ***$P<0.001$ (two-tailed two-sample unequal variance Student *t*-test). (**g**) Thy1.1$^+$ OT-II cells infected with control (shNC) or *Etv5* shRNA (shETV5) expressing retrovirus were subjected to adoptive transfer experiment. Eight days after immunization, donor cells were analysed for T$_{FH}$ cell differentiation. shNC, $n=6$; shETV5, $n=8$. **$P<0.01$ (two-tailed two-sample unequal variance Student *t*-test). (**h**) WT and *Cic* cKO OT-II cells were prepared from spleens of Thy1.1$^+$ OT-II *Cic*$^{f/f}$ and Thy1.1$^+$ OT-II *Cic*$^{f/f}$*Cd4-Cre* mice, respectively. WT OT-II cells infected with shNC retrovirus and *Cic* cKO OT-II cells infected with shNC or shETV5-expressing retrovirus were transferred into Thy1.2$^+$ B6 recipient mice. Seven days after immunization with NP-OVA in alum, donor cells were analysed for T$_{FH}$ cell differentiation. WT:shNC, $n=9$; cKO:shNC, $n=6$; cKO:shETV5, $n=6$. ***$P<0.001$ (two-tailed two-sample unequal variance Student *t*-test).

**Figure 4 | Spontaneous induction of T$_{FH}$ cell differentiation and GC response in T-cell-specific *Cic* null mice.** (**a**) Western blot analysis showing T-cell-specific ablation of CIC expression in *Cic*$^{f/f}$*Cd4-Cre* mice. B220$^+$ B and CD4$^+$ T cells were purified from spleen of 12-week-old *Cic*$^{f/f}$ (WT) and *Cic*$^{f/f}$*Cd4-Cre* (cKO) mice. (**b**) Levels of total IgG, IgG1, IgG2a and IgM in sera from 12-week-old *Cic*$^{f/f}$ and *Cic*$^{f/f}$*Cd4-Cre* mice. (**c**) Serum levels of anti-dsDNA antibody in 14.5-month-old *Cic*$^{f/f}$ and *Cic*$^{f/f}$*Cd4-Cre* mice were measured by ELISA. (**b,c**) The graphs show data as mean ± s.e.m. and each dot in graphs represent an individual mouse. *$P<0.05$ and **$P<0.01$ (two-tailed two-sample unequal variance Student *t*-test). (**d**) Immune cell infiltration in liver and lung from 14.5-month-old *Cic*$^{f/f}$*Cd4-Cre* mice. Tissue sections were stained with H&E. (**e**) H&E-stained kidney sections showing glomerulonephritis in 14.5-month-old *Cic*$^{f/f}$*Cd4-Cre* mice. (**d,e**) Representative images from two *Cic*$^{f/f}$*Cd4-Cre* mice (cKO-1 and cKO-2) are shown. Scale bars, 100 μm. (**f–i**) Flow cytometry analyses of non-T$_{FH}$, T$_{FH}$ and GC T$_{FH}$ cells (**f**), CD4$^+$ICOS$^+$ T cells (**g**), GC B cells (**h**) and CD4$^+$FOXP3$^-$PD-1$^+$CXCR5$^+$ T$_{FH}$ and CD4$^+$FOXP3$^+$PD-1$^+$CXCR5$^+$ T$_{FR}$ cells (**i**) in spleen from 12-week-old *Cic*$^{f/f}$ and *Cic*$^{f/f}$*Cd4-Cre* mice. All data are representative of three independent experiments with $n=4–5$ mice per group in each experiment. Numbers adjacent to outlined areas indicate per cent of each cell population among splenic CD4$^+$ or B220$^+$ cells. *$P<0.05$, **$P<0.01$ and ***$P<0.001$ (two-tailed two-sample unequal variance Student *t*-test). (**j**) Quantification of the ratio T$_{FH}$/T$_{FR}$ from the experiments as in Fig. 3i. **$P<0.01$ (two-tailed two-sample unequal variance Student *t*-test).

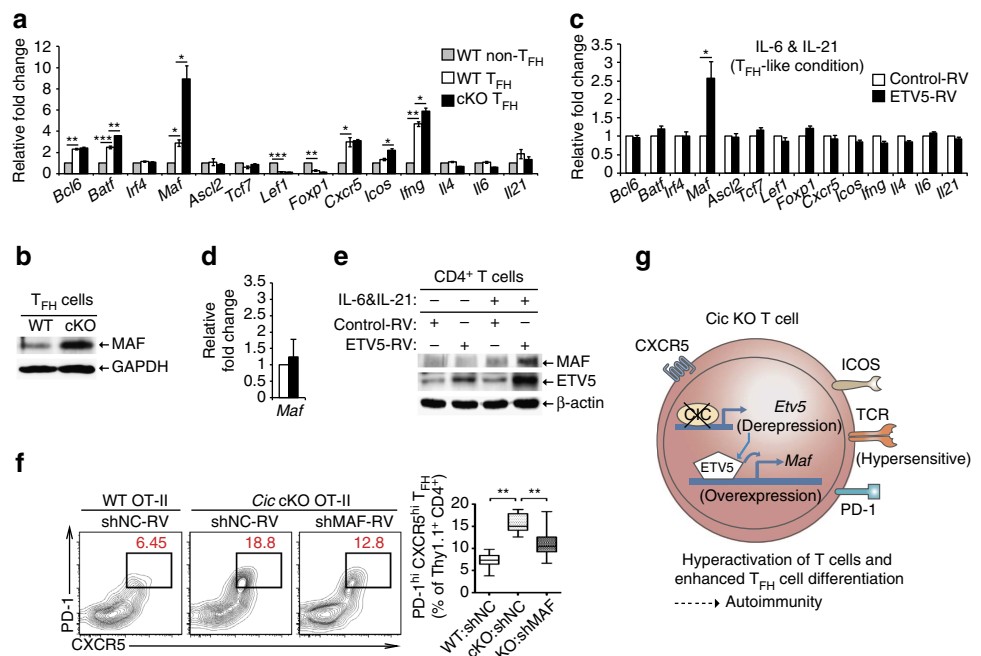

**Figure 6 | *Maf* is a critical downstream target of the CIC–ETV5 axis in the process of T$_{FH}$ cell differentiation.** (**a**) qRT-PCR analysis for levels of T$_{FH}$-related genes in sorted non-T$_{FH}$ and T$_{FH}$ cells from *Cic*$^{f/f}$ and *Cic*$^{f/f}$*Cd4-Cre* mice. Splenocytes from 5 to 6 *Cic*$^{f/f}$ mice or 3 to 4 *Cic*$^{f/f}$*Cd4-Cre* mice were pooled and subjected to cell sorting. Three independent experiments were performed. (**b**) FACS-sorted splenic T$_{FH}$ cells from *Cic*$^{f/f}$ and *Cic*$^{f/f}$*Cd4-Cre* mice were subjected to western blot analysis for MAF expression. The images are representative of two independent experiments. (**c**) qRT-PCR analysis of T$_{FH}$-related genes in CD4$^+$ T cells infected with control or ETV5-expressing retrovirus. The cells were incubated in the presence of IL-6 and IL-21, and re-stimulated with anti-CD3 for 2 h prior to RNA extraction. Four independent experiments were performed. (**d**) qRT-PCR analysis for *Maf* levels in CD4$^+$ T cells infected with control or ETV5-expressing retrovirus in the absence of IL-6 and IL-21. Before RNA extraction, the cells were activated with anti-CD3 for 2 h. Four independent experiments were performed. (**a,c,d**) Error bars indicate s.e.m. *$P < 0.05$, **$P < 0.01$ and ***$P < 0.001$ (two-tailed two-sample unequal variance student *t*-test). (**e**) Western blot analysis for MAF levels in CD4$^+$ T cells infected with control or ETV5-expressing retrovirus. The cells were cultured in the presence or absence of IL-6 and IL-21, and then re-stimulated with anti-CD3 for 12 h. The images are representative of two independent experiments. (**f**) WT Thy1.1$^+$ OT-II cells infected with control retrovirus and *Cic* null Thy1.1$^+$ OT-II cells infected with control or *Maf* shRNA (shMAF) expressing retrovirus were transferred into Thy1.2$^+$ B6 recipient mice. Seven days after immunization with NP-OVA in alum, the Thy1.1$^+$ OT-II cells were analysed for T$_{FH}$ cell differentiation using flow cytometry. WT:shNC, $n = 11$; cKO:shNC, $n = 9$; cKO:shMAF, $n = 8$. **$P < 0.01$ (two-tailed two-sample unequal variance Student *t*-test). (**g**) Schematic illustration on how CIC deficiency induces T-cell activation, T$_{FH}$ cell differentiation and autoimmunity. CIC deficiency in T cells makes the TCR response hypersensitive, thereby inducing T-cell activation. During T$_{FH}$ cell differentiation, de-repression of *Etv5* induces *Maf* expression in *Cic* null T$_{FH}$ cells, subsequently promoting T$_{FH}$ cell differentiation. These T-cell abnormalities could contribute to autoimmunity in the *Cic*-deficient mice.

with Stattic did not affect ETV5 promoter occupancy of *Maf* (Supplementary Fig. 15c). These results indicate that STAT3 activity is required for ETV5-mediated induction of *Maf* expression, but dispensable for ETV5 binding to *Maf* promoter.

Finally, we examined whether the upregulation of MAF expression contributed to the enhanced T$_{FH}$ cell differentiation in *Cic*-deficient T cells by the adoptive transfer experiment. WT and *Cic* null OT-II cells transduced with control or *Maf* shRNA (shMAF)-expressing retrovirus were adoptively transferred to Thy1.2$^+$ recipient mice. Seven days after immunization with NP-OVA in alum, the donor OT-II cells were analysed for T$_{FH}$ cell differentiation by flow cytometry. The enhanced T$_{FH}$ cell differentiation of *Cic*-deficient OT-II cells was indeed significantly alleviated by knockdown of *Maf* (Fig. 6f, Supplementary Fig. 13b). Taken together, these data suggest that the derepression of *Etv5* increased *Maf* expression in *Cic*-deficient T$_{FH}$ cells, and subsequently promoted T$_{FH}$ cell differentiation and GC responses.

## Discussion

In this study, we examined a previously unrecognised function of CIC in the immune system. Our findings demonstrate that CIC is

a key transcriptional repressor that maintains peripheral immune homeostasis and restrains T$_{FH}$ cell differentiation, thereby suppressing autoimmunity (Fig. 6g). CIC deficiency causes lymphoproliferative autoimmunity in mice, which is accompanied by augmented T-cell responses and increased frequencies of T$_{FH}$ and GC B cells in secondary lymphoid organs. Given that excessive formation of T$_{FH}$ cells is often observed in patients with autoimmune diseases[5] and that autoimmunity in *Roquin*$^{san}$ mice is reduced when T$_{FH}$ cell development is suppressed either by removing one allele of *Bcl6*, by deleting SAP (*Sh2d1a*), or by dampening IFNγ signalling[40,60], spontaneous induction of the GC response might be a primary cause of the onset of autoimmune-like symptoms in *Cic* mutant mice.

The immune cell-specific *Cic* null mice shared many phenotypes including hyperglobulinemia, T-cell hyperactivation, accumulation of effector/memory cells with T$_H$1 and T$_H$2 phenotypes, systemic autoimmunity and increased proportions of CD4$^+$FOXP3$^+$CD25$^-$ T, T$_{FH}$ and GC B cells with the T-cell-specific *Cic* null mice, demonstrating that T-cell-intrinsic functions of CIC are crucial for maintenance of T-cell homeostasis and suppression of spontaneous induction of the GC response and autoimmunity. A specific loss of CIC in B cells, from which antigen presentation and co-stimulatory signals are

required for differentiation and maintenance of $T_{FH}$ cells[8], did not promote $T_{FH}$ cell differentiation in mice (Supplementary Fig. 16), accentuating the T-cell-intrinsic requirement of CIC in regulation of $T_{FH}$ cell development. However, most phenotypes were more prominent in the immune cell-specific *Cic* null mice than in the T-cell-specific *Cic*-deficient mice; this difference implies that T-cell-extrinsic mechanisms also contributed in part to those phenotypes. To comprehensively understand the function of CIC in maintenance of peripheral immune tolerance, functions of CIC in other types of immune cell, such as DCs and B cells, need to be further determined.

Our study provided a molecular basis of how CIC deficiency promotes $T_{FH}$ cell differentiation. We determined that ETV5 is a critical CIC target that promotes $T_{FH}$ cell differentiation. Among *Pea3* group genes, *Etv4* and *Etv5* are major CIC target genes in $CD4^+$ T cells. Although mRNA levels of both genes increased greatly in the *Cic* null $T_{FH}$ cells, only ETV5 expression increased significantly at the protein level, indicating that ETV4 levels are tightly controlled in $T_{FH}$ cells by posttranscriptional regulation. ETV5 expression also seems to be regulated at the posttranscriptional level in the *Cic* null $T_{FH}$ cells, because the increase in ETV5 expression by CIC deficiency was greater at the mRNA level ($\sim 5.5$-fold) than at the protein level ($\sim 3.4$ fold). One possible mechanism of posttranscriptional control of PEA3 group transcription factor levels in $T_{FH}$ cells is proteasomal degradation mediated by E3 ubiquitin ligase COP1 (also known as RFWD2). COP1-mediated degradation of PEA3 group transcription factors is involved in regulation of prostate cancer progression[61], β-cell insulin secretion[62] and lung-branching morphogenesis[63], but the function of COP1 in $T_{FH}$ cell development has not been determined. Because *Rfwd2* is expressed in T lymphocytes[64], one interesting follow-up would be to examine whether COP1 regulates $T_{FH}$ cell differentiation through degradation of ETV5 or other PEA3 group transcription factors in T cells.

We identified *Maf* as an important downstream target gene of the CIC–ETV5 axis in the process of $T_{FH}$ cell differentiation. To identify genes that may be regulated by ETV5 during $T_{FH}$ cell differentiation, we examined 14 genes that encode transcription factors, cytokines or cell surface molecules, all of which are critically involved in regulation of $T_{FH}$ cell development. Strikingly, ETV5 overexpression induced only *Maf* expression in $CD4^+$ T cells. The finding that $T_{FH}$ cell differentiation of the *Cic* null OT-II cells was suppressed by knockdown of *Maf* further demonstrates that the CIC–ETV5 axis regulates $T_{FH}$ cell differentiation via MAF. However, in the *Cic*-deficient OT-II cells, *Maf* RNAi suppressed $T_{FH}$ cell differentiation less efficiently than did *Etv5* RNAi; this comparison suggests that, in addition to *Maf*, other ETV5 target genes may contribute to promotion of $T_{FH}$ cell differentiation. Moreover, *Cic*-deficient $T_{FH}$ cells exhibited the enhanced expression of IFNγ, which is critical for $T_{FH}$ cell differentiation[40], but not regulated by MAF[14]. This result also implies that *Maf* may not be the only ETV5 target gene that contributes to promotion of $T_{FH}$ cell differentiation. To improve our understanding of how ETV5 promotes $T_{FH}$ cell development, studies of the function of ETV5 in $T_{FH}$ cells, such as a genome-wide identification of ETV5 targets and their function in $T_{FH}$ cells, should be conducted.

Given that ETV5 upregulated *Maf* expression only in the presence of IL-6 and IL-21, which are cytokines that induce $T_{FH}$ cell development[56], and that *Maf* levels were increased in *Cic* null $T_{FH}$ cells, but in neither naïve nor anti-CD3/CD28-activated *Cic*-deficient $CD4^+$ T cells (Supplementary Data 1), ETV5 might specifically regulate *Maf* expression during $T_{FH}$ cell development. Consistent with this inference, *Maf* levels were comparable between WT and *Etv5* deficient $T_H17$ cells[65].

However, ETV5 facilitates differentiation of $T_H17$ cells[65] and of IL-17-producing γδ effector cells[66] by activating *Il17* expression. ETV5 also promotes IL-9 production in $T_H9$ cells by binding and recruiting histone acetyltransferases to the *Il9* locus[67]. These findings, together with our data, suggest that ETV5 participates in differentiation processes of several $T_H$ subtypes by regulating a different set of genes in each $T_H$ subset.

Our study is the first demonstration that the CIC–ETV5 axis is critical for regulation of $T_{FH}$ cell development. This axis also contributes to cancer progression[22–24]; therefore, dysregulation of this regulatory axis could contribute to pathogenesis of both cancer and antibody-mediated autoimmune diseases. In this regard, CIC and ETV5 themselves or their target genes that commonly mediate pathogenesis of both diseases could be evaluated as molecular targets for treatment of both autoimmune diseases and cancers.

## Methods

**Mice.** All mice were maintained on a C57BL/6 background. Vav1-Cre (ref. 68), Cd4-Cre (ref. 69), Foxp3-YFP-Cre (ref. 51), Cd19-Cre (ref. 70) and OT-II (ref. 55) mice have been described previously. The mouse with a loxP-flanked *Cic* allele was generated by mating $Cic^{tm1a(KOMP)Wtsi}$ to Actin-FLP mice (Stock No: 003800). The $Cic^{tm1a(KOMP)Wtsi}$ allele had been obtained as ES cells (Clone No. EPD0285_3_F07) from EUCOMM and injected into C57BL/6 blastocysts. Animals were maintained in a specific pathogen-free animal facility under standard 12 h light/12 h dark cycle. Mice were fed standard rodent chow and water *ad libitum*. Three-to-thirteen mice that were used in individual experiments were assigned randomly to the experimental groups. Male and female mice were also randomly assigned to the experimental groups. Blinding was not possible in most animal experiments. All procedures were approved by the Pohang University of Science and Technology Institutional Animal Care and Use Committee.

**Enzyme-linked immunosorbent assay.** Ninety six-well ELISA plates were pre-coated with $2 \mu g\,ml^{-1}$ anti-mouse Ig (1010-01, Southern Biotechnology) at 4 °C overnight. On the next day, the plates were washed with washing buffer (phosphate-buffered saline (PBS) with 0.05% Tween 20) and blocked with blocking buffer (washing buffer with 2% bovine serum albumin (BSA)) for 1–2 h at room temperature (RT); then 50 μl of diluted serum and standards were added and incubated for 2 h at RT. The plates were washed and incubated for 1 h at RT with secondary antibodies conjugated with horseradish peroxidase. For total IgG, goat anti-mouse IgG (H + L) (pooled antisera from goats hyperimmunized with mouse IgG, 1:3,000 diluted, 1031-05) was used; for IgG1, goat anti-mouse IgG1 (pooled antisera from goats hyperimmunized with mouse IgG1, 1:3,000 diluted, 1070-05) was used; for IgG2a, goat anti-mouse IgG2a (pooled antisera from goats hyper-immunized with mouse IgG2a paraproteins, 1:1,000 diluted, 1080-05) was used; and for IgM, goat anti-mouse IgM (pooled antisera from goats hyperimmunized with mouse IgM, 1:2,000 diluted, 1020-05) was used. The plates were washed at least five times, then 50 μl of TMB substrate (SurModics, TMBM-1000-01) was added and incubated for 20 min in the dark. Finally, 50 μl of stop solution (1 M $H_2SO_4$) was added and the plate was read at 450 nm. The serum concentration was calculated according to the standard curve generated. Serum anti-dsDNA concentration was measured using mouse anti-dsDNA ELISA Kit (Shibayagi, AKRDD-061) according to the manufacturer's instruction. For IL-2 ELISA, naïve $CD4^+$ T cells were activated with plate-bound anti-CD3 ($1.0 \mu g\,ml^{-1}$, 145-2C11, BD) in the presence or absence of plate-bound anti-CD28 ($2.0 \mu g\,ml^{-1}$, 37.51, BD), then cell supernatants were collected 48 and 72 h after stimulation. Briefly, plates pre-coated with $2 \mu g\,ml^{-1}$ anti-IL-2 (14-7022, eBioscience) were washed and blocked as described above, then 50 μl of diluted samples and standards were added and incubated for 2 h at RT. The plates were washed, then incubated sequentially with biotin-conjugated anti-IL-2 (13-7021) for 1.5 h, avidin-HRP (18-4100) for 30 min, TMB substrate for 15 min, and 1 M $H_2SO_4$ as a stop solution.

**Tissue histology.** Liver, lung and kidney tissues obtained from 12-month-old $Cic^{f/f}$ and $Cic^{f/f}Vav1$-Cre mice or 14.5-month-old $Cic^{f/f}$ and $Cic^{f/f}Cd4$-Cre mice were fixed in 10% formalin, then embedded in paraffin before sectioning. The tissues were cut into 5-μm sections (Leica RM2245), then sections were deparaffinized and dehydrated by using xylene, 100% ethanol and 95% ethanol sequentially. They were washed in distilled water, then stained with haematoxylin (Sigma, HHS32) and eosin (Sigma, HT110132). After H&E staining, × 200 images of specimens were obtained using an Olympus CX31 light microscope.

**Immunofluorescence staining of IgG deposition.** Kidney tissues from 12-month-old $Cic^{f/f}$ and $Cic^{f/f}Vav1$-Cre mice were snap-frozen in OCT medium (Sakura Finetek, 4583), then cut into 10-μm sections (Leica CM3050S). The kidney sections were air-dried for 24 h, then washed in PBS (pH 7.4) to remove fixation

compound. Diluted anti-mouse CD16/CD32 antibodies (eBioscience, 14-0161) were treated for Fc blocking and incubated for 30 min at RT. After washing, the kidney sections were stained with anti-IgG antibody conjugated to fluorescein isothiocyanate (FITC) (1:500 diluted, Sigma, F0257) and 4′,6-diamidino-2-phenylindole to reveal the IgG complexes and the nuclei, respectively. The specimens were washed in PBS, then covered in mounting medium (Dako, C0563), and ×400 images of specimens were obtained using an Olympus IX82-ZDC2 fluorescence microscope.

**Plasmids and retrovirus production.** The amino-terminal FLAG-tagged coding sequence (CDS) of mouse *Etv5* was amplified by PCR, initially cloned into T-blunt vector (SolGent), and then verified by sequencing. A confirmed *Etv5* CDS fragment digested by XhoI/HpaI was sub-cloned into MigR1 retroviral vector (MigR1-ETV5-GFP). The shRNA expression vectors for knockdown of mouse *Etv5* and *Maf* were generated using MSCV-LTRmiR30-PIG (LMP) vector (Open Biosystems) according to the manufacturer's instruction. The target sequences were as follows. For shETV5: 5′-ACCCGAGAGACTGGAAGGCAAA-3′ and for shMAF: 5′-AAG ATATAACCTGCAAGCATAT-3′.

Viruses were generated by transient co-transfection of Platinum-E (Plat-E) retroviral packaging cell line (Cell Biolabs) with the cloned retroviral vectors and pCL-Eco helper plasmid (Imgenex). Briefly, $0.5–0.8 \times 10^6$ plat-E cells were plated in 6-well plates. On the next day, the cells were transfected with $1.2 \mu g$ of retroviral vector and $0.8 \mu g$ of pCL-Eco using FuGENE HD transfection reagent (E2311, Promega). Retrovirus-containing supernatants were collected 48 h later and frozen at $-80 °C$.

**Flow cytometry and cell sorting.** Single-cell suspensions of spleens and thymuses were prepared and surface-stained in FACS buffer (PBS + 1.5% fetal bovine serum (FBS)) with monoclonal antibodies. The following antibodies were obtained from eBioscience, BD PharMingen or BioLegend: anti-CD4 (1:100 diluted, GK1.5), anti-CD8 (1:100 diluted, 53-6.7), anti-CD3 (1:100 diluted, 17A2), anti-CD11b (1:100 diluted, M1/70), anti-CD11c (1:100 diluted, N418), anti-MHCII (1:100 diluted, AF6-120.1), anti-F4/80 (1:100 diluted, BM8), anti-CD44 (1:100 diluted, IM7), anti-CD62L (1:100 diluted, MEL-14), anti-CD25 (1:50 diluted, PC61), anti-PD-1 (1:50 diluted, J43), anti-GITR (1:100 diluted, DTA-1), anti-OX40 (1:100 diluted, OX-86), anti-ICOS (1:100 diluted, 7E.17G9), anti-B220 (1:200 diluted, RA3-6B2), anti-IgM (1:100 diluted, II/41), anti-CD43 (1:100 diluted, R2/60), anti-GL-7 (1:250 diluted, GL7), biotinylated anti-mouse FAS (1:200 diluted, Jo2), anti-CD40 (1:100 diluted, 1C10), anti-CD80 (1:100 diluted, 16-10A1), anti-CD86 (1:100 diluted, GL1), anti-ICOSL (1:100 diluted, HK5.3), anti-PDCA-1 (1:100 diluted, eBio927), anti-CCR4 (1:100 diluted, 2G12), anti-CCR6 (1:100 diluted, 140706), anti-CXCR3 (1:150 diluted, CXCR3-173), anti-CTLA-4 (1:100 diluted, UC10-4B9), anti-CD103 (1:100 diluted, 2E7), anti-GARP (1:100 diluted, YGIC86), anti-IgG2a (1:100 diluted, m2a-15F8), anti-CD304 (Neuropilin-1; 1:50 diluted, 3E12) and APC-conjugated streptavidin (1:100 diluted, eBioscience). For CXCR5 staining, tertiary staining was used[11]. Briefly, the cells were sequentially incubated with following reagents: rat anti-CXCR5 (1:100 diluted, 2G8) for 1 h, biotinylated anti-rat IgG (1:400 diluted, eBioscience) for 30 min, and then APC- or PerCP-Cy5.5-labelled streptavidin (1:100 diluted, eBioscience) with other surface antibodies. For intracellular staining, cells were fixed and permeabilized with 'Foxp3 staining buffer set' (00-5523, eBioscience) following the manufacturer's protocol, then stained with anti-Foxp3 (1:75 diluted, MF23, BD), anti-BCL6 (1:50 diluted, 7D1), anti-T-bet (1:100 diluted, 4B10), anti-GATA3 (1:100 diluted, TWAJ), anti-RORγt (1:100 diluted, B2D), anti-Nur77 (1:100 diluted, 12.14) and anti-HELIOS (1:100 diluted, 22F6). For cytokine staining, total splenocytes were stimulated with PMA and ionomycin in the presence of Golgi-stop (554724, BD) and Golgi-plug (555029, BD) for 5 h, and then the intracellular staining of cytokines was performed using antibodies for IFNγ (1:100 diluted, XMG1.2), IL-2 (1:100 diluted, JES6-5H4), IL-17A (1:100 diluted, TC11-18H10), TNF (1:200 diluted, MP6-XT22), IL-21 (1:100 diluted, mhalx21), IL-4 (1:50 diluted, 11B11), IL-9 (1:100 diluted, RM9A4), IL-13 (1:100 diluted, eBio13A) and IL-22 (1:100 diluted, Poly5164). The stained cell samples were analysed using either a CantoII flow cytometer (BD Biosciences) or a LSRFortessa flow cytometer (BD Biosciences). Data were analysed by FlowJo software (Tree Star). MoFlo-XDP (Beckman Coulter) was used for cell sorting. The sorted populations were >98.5% pure. In FACS plots, isotype controls for each of antibodies were used for separating negative and positive populations and all gates were based on this method. All FACS sorting/gating strategies are described in Supplementary Fig. 17.

**In vitro T-cell proliferation assay.** Naïve $CD4^+$ T cells were purified to >95% purity from pooled spleens and lymph nodes of *Cic*[f/f] and *Cic*[f/f]*Vav1*-Cre mice by using a $CD4^+$ negative selection method (Stem Cell Technologies) for $CD4^+$ $CD25^-CD44^{lo}CD62L^{hi}$ cells. The cells were labelled with $5 \mu M$ Cell Trace Violet (CTV, Invitrogen, C34557) in pre-warmed PBS. Labelling was performed by incubating in a $37 °C$ incubator for 20 min and stopped by adding five times the original staining volume of culture medium (containing ≥1% protein). The cells were incubated for at least 10 min to allow the CTV reagent to undergo acetate hydrolysis. The cells were stimulated with plate-bound anti-CD3 ($1.0 \mu g ml^{-1}$,

145-2C11, BioXcell) and anti-CD28 ($2.0 \mu g ml^{-1}$, 37.51, BioXcell), then collected 72 h after stimulation. The samples were analysed using the LSRFortessa flow cytometer. Data were analysed using FlowJo software (Tree Star).

**In vitro T-cell differentiation assay.** Naïve $CD4^+$ T cells ($CD4^+CD25^-$ $CD44^{lo}CD62L^{hi}$) were stimulated with plate-bound anti-CD3 and anti-CD28 in medium supplemented as follows: for $T_H0$ differentiation, anti-IL-4 ($10 \mu g ml^{-1}$) and anti-IFNγ ($10 \mu g ml^{-1}$); for $T_H1$ differentiation, IL-12 ($10 ng ml^{-1}$) and anti-IL-4 ($10 \mu g ml^{-1}$); for $T_H2$ differentiation, IL-4 ($10 ng ml^{-1}$) and anti-IFNγ ($10 \mu g ml^{-1}$); for $T_H17$ differentiation, IL-1β ($20 ng ml^{-1}$), IL-6 ($20 ng ml^{-1}$), TGF-β ($2 ng ml^{-1}$), anti-IFNγ ($10 \mu g ml^{-1}$) and anti-IL-4 ($10 \mu g ml^{-1}$). The samples were analysed using the LSRFortessa flow cytometer, and data were analysed using FlowJo software.

**In vitro T_reg cell suppression assay.** $CD4^+CD25^-$ $CD44^{lo}$ T cells were prepared from CD45.1 C57BL/6 mice. The cells were labelled with CTV and used as responder cells (Tresp). $CD4^+CD25^+$ $T_{reg}$ cells were isolated from *Cic*[f/f] and *Cic*[f/f]*Cd4*-Cre mice by the MoFlo-XDP (Beckman Coulter). Tresp cells were mixed with an equal number of irradiated splenocytes (3000 Rad) from *Rag1*[−/−] mice and different numbers of $CD4^+CD25^+$ $T_{reg}$ cells and incubated in round bottom 96-well tissue culture plates with anti-CD3 ($1 \mu g ml^{-1}$). Seventy-two hours later, $CD45.1^+$ T cells were gated and proliferation was analysed based on CTV dilution. For detection of IL-2 secretion, $CD4^+CD25^+$ $T_{reg}$ cells from *Cic*[f/f] and *Cic*[f/f]*Cd4*-Cre mice were used at a fixed concentration of $1 \times 10^5$ cells per well and co-cultured with $CD4^+CD25^-$ Tresp cells in 96-well plates. The cells were stimulated with plate-bound anti-CD3 ($1 \mu g ml^{-1}$) and anti-CD28 ($1 \mu g ml^{-1}$), then supernatants were collected 72 h after stimulation. The samples were subjected to ELISA for IL-2 concentration.

**Thymic negative selection assay.** In vitro thymic negative selection assay was performed as described previously[45]. Briefly, DP thymocytes were prepared from *Cic*[f/f] and *Cic*[f/f]*Cd4*-Cre mice using the MoFlo-XDP and stimulated with anti-CD3 ($20 \mu g ml^{-1}$) and anti-CD28 ($50 \mu g ml^{-1}$) for 24 h to evaluate TCR-induced apoptosis. Cell death was assessed by staining for Annexin V according to the manufacturer's protocol (BD, 556419). To investigate the induction of Nur77 expression after TCR cross-linking, thymocytes from *Cic*[f/f] and *Cic*[f/f]*Cd4*-Cre mice were stimulated with anti-CD3 ($5 \mu g ml^{-1}$) and anti-CD28 ($10 \mu g ml^{-1}$) for 6 h. The samples were analysed using the LSRFortessa flow cytometer, and data were analysed using FlowJo software.

**Western blot analysis.** Single-cell suspensions of spleen, lymph nodes and thymus from *Cic*[f/f] and *Cic*[f/f]*Vav1*-Cre mice were used as total immune cells in lymphoid organs. Naïve $CD4^+$ T cells were activated by anti-CD3 and anti-CD28 and collected on days 0 (unstimulated), 1, 2 and 3. $B220^+$ B and $CD4^+$ T cells from *Cic*[f/f] and *Cic*[f/f]*Cd4*-Cre mice were prepared using MACS (130-049-501 and 130-104-454, Miltenyi Biotec) according to the manufacturer's protocol. Naïve $CD4^+$ T, effector/memory $CD4^+$ T, $T_{FH}$ and non-$T_{FH}$ cells were purified from total $CD4^+$ T cells by FACS sorting using the MoFlo-XDP. Splenic $CD19^+$ B and $CD3^+$ T cells were prepared from *Cic*[f/f] and *Cic*[f/f]*Cd19*-Cre mice by FACS sorting using the MoFlo-XDP. Total lysates from 0.2 to $1.0 \times 10^6$ cells of each cell population were separated by sodium dodecyl sulfate polyacrylamide gel electrophoresis (SDS–PAGE). The primary antibodies used are as follows. anti-CIC (1:500 diluted, homemade)[28], anti-ETV4 (1:1,000 diluted, 10684-1-AP, ProteinTech), anti-ERM/ETV5 (1:1,000 diluted, ab102010, Abcam), anti-MAF (1:750 diluted, sc-7866, Santa Cruz Biotechnology), anti-β-actin (1:2,000 diluted, sc-47778, Santa Cruz Biotechnology) and anti-GAPDH (1:2,000 diluted, sc-32233, Santa Cruz Biotechnology). Raw western blot images are presented in Supplementary Figs 18 and 19.

**Retroviral transduction and cell transfers.** OT-II Thy1.1 $CD4^+$ T cells were purified from whole splenocytes by negative selection using EasySep (StemCell), then re-suspended in T-cell medium (RPMI-1640 + 10% FBS, supplemented with 2.05 mM L-Glutamine, 100 U ml$^{-1}$ Penicillin/Streptomycin and 55 mM BME). The purified OT-II T cells were stimulated in 24-well plates pre-coated with $2 \mu g ml^{-1}$ anti-CD3 (145-2C11, BD) and anti-CD28 (37.51, BD) in the presence of rmIL-2 ($50 U ml^{-1}$). The OT-II T cells were transduced with control retrovirus (MigR1-GFP) or ETV5-expressing retrovirus (MigR1-ETV5-GFP) at 24 and 48 h after in vitro stimulation. After a total of 72 h stimulation, the cells were transferred into new six-well plates with rmIL-2 for 24 h, and then transferred again into new six-well plates in the presence of rmIL-7 for 24 h prior to cell sorting. For shRNA experiments, after in vitro stimulation and double transduction with shNC, shETV5 or shMAF expressing retrovirus for 72 h, the cells were transferred into six-well plates with rmIL-2 for 12 h and then transferred into new six-well plates with rmIL-7 for 12 h.

For adoptive transfer experiments, $0.2–0.5 \times 10^6$ OT-II T cells infected with retroviruses were transferred into C57BL/6 Thy1.2 recipient mice by tail-vein injection. Three days later, mice were immunized intraperitoneally with $200 \mu g$ of NP-OVA (9006-59-1, Sigma) in $200 \mu l$ of alum. Seven or eight days after

immunization, the donor cells were analysed for $T_{FH}$ differentiation by flow cytometry.

**Cell culture.** Naïve ($CD4^+CD25^-CD44^{lo}CD62L^{hi}$) $CD4^+$ T cells were isolated by negative selection using EasySep (StemCell), then activated with plate-bound anti-CD3 (145-2C11, BD) and anti-CD28 (37.51, BD) under neutral (anti-IFNγ, anti-IL-4, anti-TGFβ) or $T_{FH}$-like (anti-IFNγ, anti-IL-4, anti-TGFβ, rmIL-6, rmIL-21) condition. The cells were transduced with retrovirus (MigR-GFP or MigR-ETV5-GFP) at 24 h and at 36 h after *in vitro* stimulation. Two days after culture, the cells were rested for 24 h in the presence of rmIL-7, then re-stimulated with pre-coated anti-CD3 for 2 h for RNA samples and for 12 h for protein samples.

For STAT3 inhibitor treatment experiments, naïve $CD4^+$ T cells were polarized under $T_{FH}$-like condition as described above. The cells were transduced with retrovirus (MigR1-GFP or MigR1-ETV5-GFP) at 24 h after *in vitro* stimulation, then treated with AG490 (Tokyo Chemical, 50 μM) for 16 h or Stattic (Santa Cruz Biotechnology, 20 μM) for 12 h. Two days after culture, the cells were rested for 24 h. The cells were collected for RNA samples.

**ChIP and qPCR.** Chromatin immunoprecipitation was performed as previously described[28]. Chromatin from $5 \times 10^6$ $CD4^+$ T cells were crosslinked in 1.5% formaldehyde for 12 min with constant shaking, quenched cross-linking by adding 1.5 M glycine for 10 min, and then rinsed with cold PBS twice. After centrifugation for 5 min, the pellet was resuspended in Buffer A (100 mM Tris pH 9.4, 1 × protease inhibitor cocktail (11836170001, Roche)), Buffer 1 (10 mM HEPES, 10 mM EDTA, 0.5 mM EGTA, 0.25% Triton-X, 1 × protease inhibitor cocktail), and Buffer 2 (10 mM HEPES, 0.2 M NaCl, 1 mM EDTA, 0.5 mM EGTA, 1 × protease inhibitor cocktail), successively. T-cell lysate was resuspended in 200 μl of nuclei lysis buffer (5 mM Tris pH 8.1, 10 mM EDTA, 1% SDS, 1 × protease inhibitor cocktail) and sonicated. After microcentrifugation, the supernatant was pre-cleared with protein G agarose (16-266, Millipore) in dilution buffer (16.7 mM Tris pH 8.1, 167 mM NaCl, 1.2 mM EDTA, 0.01% SDS, 1.1% Triton-X, 1 × protease inhibitor) for 1–2 h. Six micrograms of anti-CIC antibody (homemade)[28] were added to chromatin samples and incubated overnight at 4 °C. The chromatin and antibody mixtures were further incubated with protein G agarose for 2–4 h at 4 °C. After washing, bound chromatins were eluted twice by elution buffer (0.5% SDS and 0.1 M NaHCO$_3$) and reverse-crosslinked with 200 mM NaCl for at least 4 h at 65 °C. Proteins were digested by proteinase K and DNA was purified by AccuPrep Gel Purification Kit (K-3034, Bioneer). qPCR was performed to quantify the relative enrichment of promoter regions of *Pea3* group genes in the immunoprecipitated DNA fragments.

For ETV5 ChIP experiment, $CD4^+$ T cells were activated with plate bound anti-CD3 (145-2C11, BD) and anti-CD28 (37.51, BD) in the presence of rmIL-6. Sixteen hours after *in vitro* stimulation, $1 \times 10^7$ cells and 2 μg of anti-ETV5 antibody (sc-22807, Santa Cruz Biotechnology) were used for each ChIP experiment. To inhibit STAT3 activity, $CD4^+$ T cells were activated with anti-CD3/CD28 in the presence of rmIL-6 for 12 h, and then Stattic (20 μM) was treated for 12 h. The primer sequences used are listed in Supplementary Data 2.

**qRT-PCR.** Total RNA was extracted using Trizol reagent (301-001, GeneAll), then 1–2 μg of the total RNA was subjected to cDNA synthesis using a GoScript Reverse Transcription System (A5000, Promega). Each gene expression level was normalized to *Hprt* levels and presented as relative to WT. The primers used in qRT-PCR analyses are listed in Supplementary Data 3.

**RNA sequencing and data analysis.** Spleens were dissected from 10-week-old $Cic^{f/f}$ and $Cic^{f/f}Cd4$-Cre mice. $CD4^+$ T cells were pre-enriched using EasySep (19860 and 18001, StemCell), then further sorted on the basis of surface markers of naïve $CD4^+$ T cells ($CD4^+CD44^{lo}CD62L^{hi}$). Sorted naïve $CD4^+$ T cells were activated in 24-well plates pre-coated with 2 μg ml$^{-1}$ anti-CD3 (145-2C11, BD) and anti-CD28 (37.51, BD) for 3d. Total RNA was extracted from naïve and activated $CD4^+$ T cells using ReliaPrep RNA Cell Miniprep Kit (Z6011, Promega). The library for mRNA sequencing was generated using a Illumnia TruSeq Preparation Kit (RS-122-2001) and sequenced on a NextSeq 500 sequencer (Illumnia). Tophat (v 2.0.10) was used to map sequencing reads to the mouse reference genome (mm9 RefSeq). Assembly of transcripts and identification of differentially expressed genes (DEGs, Fold change ($\log_2$) > 1.5; $P < 0.05$) were conducted by Cufflinks (v 2.1.1).

**Statistical analysis.** For statistical analysis, all experiments were performed more than three times independently. Statistical analyses were performed using the Student *t*-test (two-tailed, two-sample unequal variance). $P < 0.05$ was considered significant. In bar graphs, bars indicate means and error bars indicate SEM. In box-and-whisker plots, boxes represent median with upper and lower quantiles and whiskers represent values of maximum and minimum.

**Data availability.** The authors declare that the data supporting the findings of this study are available within the article and its Supplementary Information Files,

or from the corresponding authors on reasonable request. The RNA-seq data were deposited in the Gene Expression Omnibus (NCBI) data repository under accession number GSE84125.

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

## Acknowledgements

We thank Dr Huda Y. Zoghbi for provision of the Cic floxed mice, Drs Alexander Y. Rudensky and Dipayan Rudra for provision of Foxp3-YFP-Cre mice, and Drs Yeonseok Chung, V. Narry Kim and Charles D. Surh for helpful discussions and comments on this study. This work was supported by Samsung Science and Technology Foundation under project number SSTF-BA1502-14 to Y.L. S.P., S.L., H.H., J.-S.L., Y.M.K. and G.Y.P. were supported by the BK21 Plus Program. S.-W.L. was supported by the grant from Cooperative Research Program for Agriculture Science and Technology Development under project number PJ01131603 (Rural Development Administration, Republic of Korea). S.-H.I. was supported by the grant from the Institute for Basic Science (IBS-R005-G1). D.H. was supported by the grant from the Institute for Basic Science (IBS-R013-G1).

## Author contributions

Y.L., S.-W.L., S.-H.I., S.P., S.L. and C.-G.L. designed the study. S.P., S.L., C.-G.L., H.H., J.-S.L., Y.M.K. and G.Y.P. performed the experiments. D.H. analysed RNA sequencing data. S.B.L., Y.S.C. and J.D.F. provided intellectual contributions. Y.L., S.-W.L. and S.P. wrote the paper. S.-H.I., Y.S.C. and J.D.F. edited the paper.

## Additional information

Competing interests: The authors declare no competing financial interests.

