## [Peer Review File · Nature Communications]

Reviewers' comments:

Reviewer #1 (Remarks to the Author):

In the manuscript entitled "Transcriptional repressor Capicua restrains follicular helper T cell differentiation and autoimmunity" the authors demonstrated that CIC deficiency in T cells leads to excessive Tfh cell development. Mechanistically, CIC controls Tfh cells development by repressing expression of ETV5 and ETV5-target molecule cMaf. This study is important for understanding of mechanisms controlling humoral immune responses. In addition CIC function in immune cells is totally new; however, CIC-ETV5-c-maf axis in T cells is not solely responsible for excessive Tfh development in the absence of CIC. Additional molecular as well as cellular mechanisms (potential contribution of B cells and CD8+ regulatory T cells) have to be considered as well. In this regard, further studies are required to uncover the molecular and cellular mechanisms responsible for autoimmune phenotype of CIC KO mice.

Following are the concerns:

- In figure 1-2, authors have demonstrated that deletion of CIC in hematopoietic cells associates with development of autoimmunity at 12 weeks of age. Immune cell analysis indicated that autoimmune phenotype could be due to spontaneous induction of Tfh cell development and GC responses. These conclusions led to study of intrinsic CIC requirement for Tfh cells. However, CIC is highly expressed in CD8+ T cells (Sup. Fig. 1) and in figure 1, CIC KO mice have significantly low percentage of CD8+ T cells, suggesting that CIC is required for development and function of CD8+ T cells, particularly CD8+ regulatory T cells that control Tfh cell development. The authors have to consider and study this mechanism.
- The authors never looked if suppressive function of CD4+ regulatory T cells was affected by the absence of CIC.
 - In addition to CD8+ T cells, CIC KO mice have significant enhancement in percentage of B cells (Fig. 1). B cells are required for Tfh cell development; thus it is possible that spontaneous GC responses in CIC KO mice could be due to B cells as well. The authors have to study whether intrinsic function of CIC in B cells is responsible for spontaneous development of Tfh cells and autoimmunity in CIC KO mice.
- The authors stated that overexpression of IFN γ and ICOS by CIC KO T cells promotes Tfh cell development. I am wondering why expression of Tfh-specific cytokines such as IL-21 and IL-4 by CIC KO T cells was not measured. In addition to characterization of transcriptional factors (Sup Fig. 4), I recommend to examine cytokine profile in in vitro differentiated WT and CIC KO Th cells.
 - The authors indicate that CIC KO CD4+ T cells secreted more IL-2 compared to WT cells. It is known that IL-2/STAT5 pathway suppresses Tfh cell development. In this contest how can authors explain high spontaneous Tfh cell development in the presence of enhanced IL-2 production and potentially IL-2 signaling.
- Authors have to present the evidence that c-Maf is a direct target of ETV5.
 - It is interesting that in Fig. 6e, ETV5 could induce c-Maf expression in T cells only in the presence of IL-6 and IL-21, indicating that ETV5 alone is insufficient to induce c-Maf expression and additional signaling components are required and has to be determined.
 - Knockdown of c-Maf in CIC KO T cells (Fig. 6f) only moderately affected Tfh cell development, further suggesting that ETV5-c-Maf axis is not indispensable and unique for Tfh cell development in the absence of CIC. Other ETV5 targets (Batf, ICOS or others) have to be considered and examined.

Reviewer #2 (Remarks to the Author):

In this manuscript Park et al investigate the role of the transcriptional repressor Capicua (CIC) in the immune system with particular emphasis on follicular helper T cell (Tfh) development. They revealed Cic deficient mice have an increase in B cells and both CD4+ and CD8+ T cells display an activated phenotype. Compared to WT mice, Cic KO mice have hyperglobulinemia and signs of autoimmunity including enlarged secondary lymphoid organs, increases serum anti-dsDNA Ab, immune cell infiltration in the tissues, glomerulonephritis, and deposition of IgG in the kidney glomeruli. They also observed an increase in IFN γ + CD4+ T cells, FoxP3+CD25- CD4+ T cells and Tfh cells, the later of which may be related to an increase in GC B cells. A similar increase in Tfh cells, GC B cells, and hyperglobulinemia was also observed in CD4-specific Cic KO mice, indicating these defects were CD4+ T cell intrinsic. They go on to show the Pea3 group genes Etv1, Etv4 and Etv5 are direct targets of CIC and that ETV5 is specifically up-regulated in Tfh cells from Cic KO mice. Finally the Tfh cell associated transcription factor c-maf was found to be induced by ETV5. Their working model is that CIC normally represses Etv5, however in the absence of Cic, ETV5 is up-regulated, which in turn results in overexpression of c-maf and Tfh cell formation and subsequent autoimmunity.

Overall, this was a well-written manuscript, which was easy to follow; however, this reviewer is not convinced Cic acts directly on Tfh cell formation and not general T cell activation. Indeed they showed both CD4+ and CD8+ T cells are activated in the absence of Cic as well as other CD4+ T cell subsets such as FoxP3+CD25- CD4+ T cells. The increase in IFN γ and TNF α would suggest that Th1 cells are also increased. The authors should investigate other T helper cells, Th1, Th2, Th17 etc. Is expression of CXCR3, CCR6 and CCR4 also increased? Similarly is there an increase in IL-4, IL-13, IL-9 or Th17 cytokines other than IL-17A? ie IL-17F, IL-22? This is important, and as the authors point out in their discussion, ETV5 has been implicated in promoting the differentiation of other T helper subsets such as Th17 and Th9 cells. Is Cic having a general affect on CD4+ T cell activation and differentiation? What about activation markers? Along these same lines: Are the CXCR5+PD1hi cells in the absence of Cic functional Tfh cells or just activated CD4+ T cells? Do these cells make IL-21 and provide help following adoptive transfer experiments, as they don't seem to express Bcl6. Another point: are the Foxp3+CD25- CD4+ T cells, which are presumably Tregs, functional?

Other comments:

- There was no mention of the decrease in total CD8+ T cells in Cic KO mice?
- The gates in Figure 1c need to extend to the axis to include the whole population – this may look better if you use bi-exponential transformation of your data.
 - Why was the CD65L+CD44+ CD8+ effector memory population not included in the analysis although they were gated?
- In numerous FACS plots it is not evident how the gate was determined. For example Figure 2b, c, Figure 4d what was this gate based on? In particular it would be more informative to look at Bcl6 expression within the CXCR5+PD1hi population.
- Are CXCR5+PD1hi cells in Cic KO secreting IL-21 or IFN γ ?
- Line 102: they comment that CIC is not a T helper subset-specifying factor but IFN γ and TNF α are increased as well as FoxP3+CD25- CD4+ T cells suggesting CIC is having an affect on Th1 cells and Tregs.
 - Line 146: the statement “the absence of CIC relieves the normal requirement for CD28 co-stimulation” is not correct as proliferation was still increased in Cic KO compared to WT T cells in absence of anti-CD28.
- Why was addition of IL-6 and IL-21 ie Tfh-inducing conditions required to see c-maf expression? Why is Etv5 overexpression not enough?
- Why were other markers of Tfh cells such as Bcl6, Il21 and Cxcr5 not increased on Cic KO Tfh cells (Figure 6a)?
- Supplementary Figure 4: need to include a Th0 culture to determine the background in Cic KO cells. What about IL-4, IFN γ , IL-17 under these conditions? What happens when you culture under Tfh

conditions (IL-6 and IL-21) and look at Bcl6 and IL-21 expression in CIC KO compared to WT CD4 T cells? It may be more informative to include MFI of the expression of the transcription factors. How many times was this experiment performed? n = ?.

Reviewer #3 (Remarks to the Author):

The HMG-box protein Capicua (CIC) functions as a repressor of genes downstream of receptor tyrosine kinase signaling, keeping them silent in the absence of signaling. However, its function in the immune system largely remains unknown. The manuscript "Transcriptional repressor Capicua restrains follicular helper T cell differentiation and autoimmunity" by Park S et al demonstrates the autoimmune phenotype of mice with specific deletion of CIC in the immune system. The authors also observed excessive Tfh cell differentiation in these mice as well as in mice with T-cell specific deletion of CIC. They propose a model by which CIC represses the expression of Etv5 to limit the expression of cMaf expression for a normal Tfh differentiation. Many of the results are novel, which provide new insights of T-cell tolerance and Tfh differentiation. However, some aspect of the study remains incomplete and several conclusions require being refined.

Major questions:

- 1) According to the data, Vav1-cre mice demonstrated more profound immune activation than that of Cd4-cre mice. Unfortunately, it was not clearly whether Cd4-cre mice also developed tissue inflammation and systemic autoimmunity as Vav1-cre mice did. What is the exact mechanism for CIC to prevent autoimmunity? How about the innate immune system without CIC?
- 2) Is the aberrant Tfh differentiation primarily responsible for the Cd4 hyperactivation? This could be evaluated by crossing Cd4-cre mice with certain knockout mice, such as Icos KO, SAP KO or cMaf KO. The authors show CIC-deficiency affects TCR signalling. Will it affect the negative selection in the thymus? Defective negative selection could also lead to more Tfh differentiation in the peripheral but it becomes the second effect. It was striking that majority of Treg cells were CD25⁻ in the absence of CIC. Have the function of Treg cells been examined?
- 3) There were no data to show the expression levels of overexpression or knockdown of ETV5 and cMaf. Without knowing the relevance of these manipulations as compared to the expression levels of WT and CIC-deficient T cells, it is difficult to judge whether such results are physiological.

Minor questions:

- 1) The authors concluded increased B cells were responsible to splenomegaly and lymphadenopathy. How about the numbers of myeloid cells?
- 2) The authors concluded there was an increased Th1 differentiation. According to Fig 1C, almost a 2-fold increase of CD44⁺ effector/memory CD4 T cells but in Fig S3C, the increase of CD44⁺IFN γ ⁺ were much less than 2-fold. I would conclude actually there was less Th1 differentiation in total effector CD4 T cells in CIC-deficient mice.

Reviewers' comments:

Reviewer #1 (Remarks to the Author):

Revised manuscript is significantly improved; however a couple of raised concerns were not addressed in full. Thus, the following issues have to be clarified prior to consideration for publication:

1. It is still unclear whether Tregs contribute to the phenotype observed in CICf/f-Vav1Cre or CIC-CD4Cre mice. The authors indicated that the increased GC responses CIC KO mice is not due to defect in Treg cells, since suppressive activity of CIC KO nTregs towards conventional CD4⁺CD25⁻ T cells is comparable to WT Tregs. However, the role of CIC in generation and function of inducible Tregs and Tfrs was not considered. It is possible that CIC regulates suppressive function of Tregs towards specific T helper subsets. On other hand, accumulation of Tfh cells even in the present of Tfrs indicates that CIC KO Tfh (or conventional CD4⁺ T cells) could be resistant to Treg suppression. The above possibilities have to be addressed in order to clarify whether phenotype in CICf/f-CD4-Cre mice is Tregs dependent or independent.

2. Regarding mechanism, it is still unclear how ETV5/c-Maf axis could control Tfh generation. Since c-Maf is required for IL-4 and IL-21 production it is possible that ETV5/c-Maf crosstalk contributes to Tfh cell maintenance and function rather to Tfh cell priming. This is probably why knockdown of c-Maf in CIC KO T cells (Fig. 6f) only moderately affected Tfh cell priming. Authors have to extend analysis to day 14 and 21 to define if it required for Tfh maintenance.

3. Data in Fig. 6 E are not clear and contradict previous reports. Based on the data neither IL-6/IL-21 treatment nor ETV5 over-expression alone could induce c-Maf expression and presence of both components is required for c-Maf induction. However, it has been reported before that IL-6/IL-21 are potent c-Maf inducers. It is unclear why authors have different result.

4. Authors claimed that STAT3 is required for ETV5 mediated expression of c-Maf. It will be interesting to further define the nature of this cooperation and determine whether STAT3 is required for ETV5 binding to c-Maf promoter.

5. Knockdown of ETV5 in CIC KO T cells (Fig. 6f) have significant impact on Tfh cell priming, suggesting that ETV5 has additional and distinct targets from c-Maf facilitating Tfh cell priming. This target(s) have to be defined to confirm that ETV5 drives Tfh differentiation in CIC KO mice.

6. It will be interesting to examine expression of T helper cytokines in Tfh and non Tfh cells in CICf/f-CD4Cre mice. Authors for some reasons more extensively analyzed CICf/f-Vav1Cre mice compared to CICf/f-CD4Cre mice. Based on chemokine expression, CIC has intrinsic function in controlling Th1 and Th2 responses. In this regard, it is not clear why CIC does not impact The Th1 and Th2 development in vitro.

Reviewer #2 (Remarks to the Author):

Park et al have attempted to answer all the issues in their original submission. While I believe the manuscript is much improved I remain unconvinced on a few of their key findings.

- The majority of Tregs from Cic-deficient mice are CD4⁺Foxp3⁺CD25⁻, but they have yet to prove these cells in Cic-deficient mice are functional and can suppress and are thus not contributing to T cell activation and autoimmunity in Cic-deficient mice. Only CD4⁺CD25⁺ and not Foxp3⁺CD25⁻ CD4⁺ T cells were investigated in the elegantly laid out suppression assays in suppl Fig 8. This is an important point as a mechanism of suppression by Foxp3⁺CD25⁺ Tregs cells is consumption of IL-2, but this would not occur in the absence of CD25 expression ie in Foxp3⁺CD25⁻ Tregs and these form the majority of Treg compartment in Cic-deficient mice. Thus the question of whether CD4⁺Foxp3⁺CD25⁻ cells contribute to disease in Cic-deficient mice remains unanswered.

- The argument that CIC is specifically affecting Tfh cells and not other T helper cell subsets is misleading. Loss of CIC results in general T cell activation with an increase in effector and memory CD4+ T cells. While in vitro differentiation experiments of naïve cells show no difference in polarization to Th1, Th2 and Th17 cells, this was not shown for Tfh cell differentiation conditions. Furthermore, it is well known that other T helper cell subsets such as Th1, Th2, Th17 cells can also contribute to autoimmunity and/or B cell responses. Hence the title and the focus of the manuscript suggesting that Capicua specifically restrains Tfh cell differentiation is somewhat unfounded.
- The authors performed additional experiments to find comparable expression of IL-21 and IL-4 between Cic-deficient and WT Tfh cells, but significantly more IFN γ expression by Cic-deficient Tfh cells. However, the question of functional differences in driving B cell responses between Cic-deficient and WT Tfh cells still remain unanswered.

Reviewer #3 (Remarks to the Author):

The authors have addressed my questions. I recommend for publications

REVIEWERS' COMMENTS:

Reviewer #1 (Remarks to the Author):

The revised manuscript is significantly improved and the authors have addressed almost all the concerns. However, it is likely that ETV5 has other targets in addition to c-Maf. Firstly, c-maf knockdown in CIC KO T cells partially reduced Tfh cell development. In this regard, we recommend the authors to clarify that c-maf is one is potential targets of ETV5. Secondly, authors detected enhanced level of IFN γ in CIC KO Tfh cells compared to WT Tfh cells whereas IL-21 and IL-4 expression was intact. C-Maf is required for IL-4 and IL-21 production but not IFN γ . Thus, CIC or ETV5 may be additional targets responsible for IFN γ expression. This point has to be discussed in the paper.

Reviewer #2 (Remarks to the Author):

Park et al have attempted to address the remaining concerns that were raised from the revised version of this manuscript. I still feel the role of capicua is to restrain general CD4+ T cell activation and in its absence the autoimmunity present is due to overall CD4+ T cell activation, but no doubt Tfh cells are playing a role in addition to potential other CD4+ helper T cells such as Th1, Th2, and Th17 cells.

** See Nature Research's author and referees' website at www.nature.com/authors for information about policies, services and author benefits

Point-by-point response: comments made by reviewers are in **boxes** for ease of reference.

Reviewer #1:

We thank the reviewer for many constructive suggestions. We feel that the impact of our manuscript has been significantly raised by the suggested experiments.

1. Suggestion to analyze CD8⁺ regulatory T cells in *Cic* KO mice.

In figure 1-2, authors have demonstrated that deletion of CIC in hematopoietic cells associates with development of autoimmunity at 12 weeks of age. Immune cell analysis indicated that autoimmune phenotype could be due to spontaneous induction of Tfh cell development and GC responses. These conclusions led to study of intrinsic CIC requirement for Tfh cells. However, CIC is highly expressed in CD8⁺ T cells (Sup. Fig. 1) and in figure 1, *Cic* KO mice have significantly low percentage of CD8⁺ T cells, suggesting that CIC is required for development and function of CD8⁺ T cells, particularly CD8⁺ regulatory T cells that control Tfh cell development. The authors have to consider and study this mechanism.

Accordingly, we analyzed CD8⁺CD25⁺FOXP3⁺ and CD8⁺ICOSL⁺CXCR5⁺ (Qa-1-restricted CD8⁺ T cells) cells^{1,2} in spleen of WT (*Cic*^{fl/fl}) and immune cell-specific *Cic* null (*Cic*^{fl/fl}*Vav1*-Cre) mice. The results are presented in **Supplementary Fig. 8e and f**. We found that the numbers of these CD8⁺ regulatory T cells were comparable between WT and *Cic*^{fl/fl}*Vav1*-Cre mice. Therefore, we concluded that CIC is dispensable for development of CD8⁺ regulatory T cells and that the increased frequency of T_{FH} cells in *Cic*^{fl/fl}*Vav1*-Cre mice was not due to deficiency of CD8⁺ regulatory T cells.

2. Suggestion to analyze suppressive function of *Cic* deficient CD4⁺ regulatory T cells.

The authors never looked if suppressive function of CD4⁺ regulatory T cells was affected by the absence of CIC.

Accordingly, we carried out *in vitro* Treg cell suppression assay by co-culturing conventional CD4⁺CD25⁻ T cells with CD4⁺CD25⁺ Treg cells isolated from WT and *Cic* deficient mice. We determined Treg cell suppressive activity by measuring IL-2 levels in culture media and proliferation rate of the responder cells. Both WT and *Cic* deficient CD4⁺CD25⁺ Treg cells suppressed IL-2 secretion and proliferation of the responder CD4⁺ T cells with comparable efficiency. We also examined expression profiles of surface molecules, such as GITR, CTLA-4, CD103 and GARP, on CD25⁺FOXP3⁺ and CD25⁻FOXP3⁺ CD4⁺ Treg cells from WT and *Cic* deficient mice and found that there was no difference between WT and *Cic* null cells. These data are presented in **Supplementary Fig. 8b, c and d**. Based on the data, we concluded that the increased germinal center (GC) response in *Cic* deficient mice was not due to defects in CD4⁺ Treg cell function.

3. Suggestion to investigate B cell intrinsic requirement of CIC for regulation of T_{FH} cell differentiation.

In addition to CD8+ T cells, CIC KO mice have significant enhancement in percentage of B cells (Fig. 1). B cells are required for Tfh cell development; thus it is possible that spontaneous GC responses in CIC KO mice could be due to B cells as well. The authors have to study whether intrinsic function of CIC in B cells is responsible for spontaneous development of Tfh cells and autoimmunity in CIC KO mice.

This is an excellent suggestion. To address this question, we generated B cell-specific *Cic* null (*Cic^{fl/fl} Cd19-Cre*) mice and analyzed T_{FH} cells in spleen of WT and *Cic^{fl/fl} Cd19-Cre* mice at 12 weeks of age. We found that the proportion of T_{FH} cells was comparable between WT and *Cic^{fl/fl} Cd19-Cre* mice. We presented this result in **Supplementary Fig. 14**. This finding not only indicates that B cell-intrinsic function of CIC is not involved in regulation of T_{FH} cell development, but also underlines T cell-intrinsic requirement of CIC for this process.

4. Suggestion to examine levels of IL-21 and IL-4 in *Cic* null CD4⁺ T cells.

The authors stated that overexpression of IFN γ and ICOS by CIC KO T cells promotes Tfh cell development. I am wondering why expression of Tfh-specific cytokines such as IL-21 and IL-4 by CIC KO T cells was not measured.

As suggested, we carried out flow cytometry analysis on expression profiles of various cytokines including IL-21 and IL-4 in WT and *Cic* null CD4⁺ T cells. The results are presented in **Supplementary Fig. 6a**. We also determined the frequency of IL-21⁺, IL-4⁺ and IFN γ ⁺ T_{FH} cells in WT and *Cic* deficient mice (**Fig. 2e**).

5. Suggestion to examine cytokine profiles in *in vitro* differentiated WT and *Cic* null T helper cells.

In addition to characterization of transcriptional factors (Sup Fig. 4), I recommend to examine cytokine profile in *in vitro* differentiated WT and CIC KO Th cells.

We thank the reviewer for this recommendation. We examined expression profiles of IFN γ , IL-4 and IL-17A in WT and *Cic* null CD4⁺ T cells differentiated under Th0, Th1, Th2 or Th17-polarizing conditions. Bar graphs for the proportion of each cytokine-expressing T cells are included in **Supplementary Fig. 7a**.

6. Question about relationship between the increased IL-2 production and the enhanced T_{FH} cell differentiation in *Cic* deficient CD4⁺ T cells.

The authors indicate that CIC KO CD4⁺ T cells secreted more IL-2 compared to WT cells. It is known that IL-2/STAT5 pathway suppresses Tfh cell development. In this context how can authors explain high spontaneous Tfh cell development in the presence of enhanced IL-2 production and potentially IL-2 signaling.

As the reviewer pointed out, we have shown that *Cic* null CD4⁺ T cells secreted more IL-2 compared to WT cells when stimulated with anti-CD3 *in vitro* (**Fig. 3c**). This result suggests that CIC negatively regulates TCR-induced activation of T cells, at least in production of IL-2. By contrast, we have observed that the frequency of CD4⁺CD44⁺IL-2⁺ T cells was comparable between WT and *Cic*^{fl/fl}*Vav1*-Cre mice (**Supplementary Fig. 6a**). This discrepancy may stem from different T cell stimulation environments; *in vitro* activation with minimal stimuli versus *in vivo* with complex networks of co-stimulatory signals. Therefore, it is not likely that *Cic* null CD4⁺ T cells have enhanced IL-2/STAT5 signaling pathway *in vivo*.

7. Suggestion to validate that *c-Maf* is a direct target of ETV5.

Authors have to present the evidence that c-Maf is a direct target of ETV5.

It has been already reported that *c-Maf* is a direct target of ETV5 in ocular lens cells³. To confirm it, we performed chromatin immunoprecipitation in IL-6-stimulated CD4⁺ T cells using anti-ETV5 antibody followed by qPCR for two distinct *c-Maf* promoter regions containing experimentally validated ETV5 binding sites³. We found that ETV5 is indeed physically associated with *c-Maf* promoter in CD4⁺ T cells and included this result in **Supplementary Fig. 12**.

8. Suggestion to identify additional signaling components required for ETV5-mediated induction of *c-Maf* expression in CD4⁺ T cells treated with IL-6 and IL-21.

It is interesting that in Fig. 6e, ETV5 could induce c-Maf expression in T cells only in the presence of IL-6 and IL-21, indicating that ETV5 alone is insufficient to induce c-Maf expression and additional signaling components are required and has to be determined.

We agree with the reviewer that there should be additional signaling components that, in conjunction with ETV5, mediate induction of *c-Maf* expression in T cells treated with IL-6 and IL-21. Because both IL-6 and IL-21 are well known to activate STAT3 pathway, we examined whether STAT3 activation is required for the induction of *c-Maf* expression by ETV5. Treatment with chemicals that inhibit JAK2-STAT3 pathway (AG490 and Stattic) completely blocked the induction of *c-Maf* expression by ETV5 in CD4⁺ T cells in the presence of IL-6 and IL-21 (**Supplementary Fig. 13**). This result suggests that ETV5 and JAK2-STAT3 pathway might cooperatively induce expression of *c-Maf* in CD4⁺ T cells.

9. Suggestion to examine contribution of ETV5 targets other than c-MAF to the *Cic* deficiency-induced T_{FH} cell differentiation.

Knockdown of c-Maf in CIC KO T cells (Fig. 6f) only moderately affected Tfh cell development, further suggesting that ETV5-c-Maf axis is not indispensable and unique for Tfh cell development in the absence of CIC. Other ETV5 targets (Batf, ICOS or others) have to be considered and examined.

We agree that *c-Maf* is not the only ETV5 target gene responsible for the *Cic* deficiency-mediated spontaneous induction of T_{FH} cell differentiation. In this study, however, we found that only *c-Maf* expression was substantially increased by ETV5 overexpression (**Fig. 6c**) and verified that *c-Maf* is a direct target gene of ETV5 in CD4⁺ T cells (**Supplementary Fig. 12**). Although levels of *Batf*, *Icos* and *Ifng* were significantly increased in *Cic* null T_{FH} cells compared with WT T_{FH} cells (**Fig. 6a**), they are not likely direct targets of ETV5 in T_{FH} cells. Because the functions of BATF, ICOS and IFN γ in T_{FH} cell differentiation have been very well studied⁴⁻⁸, we did not further examine their contributions to the enhanced T_{FH} cell differentiation in *Cic* deficient mice. We hope to identify other ETV5 target genes in T_{FH} cells and explore their functions during T_{FH} cell differentiation as a follow-up study of this story.

Reviewer #2:

We are grateful to the reviewer for the positive comments and helpful suggestions. We have carried out additional experiments and data analyses and amended the manuscript.

1. Suggestion to investigate formation of other T helper cells in *Cic* deficient mice.

The authors should investigate other T helper cells, Th1, Th2, Th17 etc. Is expression of CXCR3, CCR6 and CCR4 also increased? Similarly is there an increase in IL-4, IL-13, IL-9 or Th17 cytokines other than IL-17A? ie IL-17F, IL-22? This is important, and as the authors point out in their discussion, ETV5 has been implicated in promoting the differentiation of other T helper subsets such as Th17 and Th9 cells. Is *Cic* having a general affect on CD4+ T cell activation and differentiation? What about activation markers?

We thank the reviewer for this suggestion. We could achieve more comprehensive data on the role of CIC in T helper cell differentiation from the experiments that the reviewer suggested. Accordingly, we analyzed expression profiles of the suggested chemokine receptors and cytokines in CD4⁺ T cells from WT and *Cic* deficient mice. We detected alterations in the formation of other T helper subsets, especially T_H1 and T_H2 cells, in *Cic* deficient mice. The data are presented in **Supplementary Fig. 6**.

2. Question about functionality of *Cic* null T_{FH} cells.

Are the CXCR5⁺PD1^{hi} cells in the absence of *Cic* functional T_{fh} cells or just activated CD4⁺ T cells? Do these cells make IL-21 and provide help following adoptive transfer experiments, as they don't seem to express Bcl6.

To address this question, we examined the proportions of IL-21⁺, IL-4⁺ and IFN γ ⁺ cells in whole splenic PD-1⁺CXCR5⁺ T_{FH} cells from WT and *Cic*^{f/f}*Vav1-Cre* mice. The data is presented in **Fig. 2e**. We found that expression of IL-21 and IL-4 was comparable between WT and *Cic* deficient T_{FH} cells, whereas IFN γ production was increased in *Cic* null T_{FH} cells, suggesting that the *Cic* deficient T_{FH} cells might be functional.

3. Question about functionality of *Cic* deficient FOXP3⁺CD25⁻CD4⁺ T cells.

Are the Foxp3⁺CD25⁻ CD4⁺ T cells, which are presumably Tregs, functional?

Previous studies have demonstrated that the FOXP3⁺CD25⁻CD4⁺ T cells have a suppressive activity and that several surface molecules are involved in controlling homeostasis and function of Treg cells^{9,10}. To address this question, we analyzed expression profiles of the surface molecules critical for Treg cell function. GITR, CTLA-4, CD103 and GARP were comparably expressed on WT and *Cic* null FOXP3⁺CD25⁻CD4⁺ T cells (**Supplementary Fig. 8b**), suggesting that loss of CIC might not affect functionality of FOXP3⁺CD25⁻CD4⁺ T cells.

4. Suggestion to mention the decrease in the number of CD8⁺ T cells in *Cic* deficient mice.

There was no mention of the decrease in total CD8⁺ T cells in *Cic* KO mice.

We stated that the number of CD8⁺ T cells was significantly decreased in *Cic*^{f/f}*Vav1-Cre* mice in the text (**page 6, lines 9 and 10**). We also examined the number of CD8⁺ regulatory T cells in WT and *Cic*^{f/f}*Vav1-Cre* mice (**Supplementary Fig. 8e and f**).

5. Suggestion to analyze the profiles of naïve and effector/memory T cells using bi-exponential transformation.

The gates in Figure 1c need to extend to the axis to include the whole population – this may look better if you use bi-exponential transformation of your data.

We thank the reviewer for this kind suggestion. We re-analyzed the data accordingly (**Fig. 1c**).

6. Suggestion to include data on the proportion of CD62L⁺CD44⁺ CD8⁺ T cells in WT and *Cic*^{f/f}*Vav1-Cre* mice.

Why was the CD62L⁺CD44⁺ CD8⁺ effector memory population not included in the analysis although they were gated?

We included a graph for the proportion of CD62L⁺CD44⁺ CD8⁺ T cells in spleen of WT and *Cic*^{fl/fl}*Vav1*-Cre mice in **Fig. 1c**.

7. Question about how to determine the gates in FACS plots.

In numerous FACS plots it is not evident how the gate was determined. For example Figure 2b, c, Figure 4d what was this gate based on?

We stained the cells using isotype controls for each of antibodies as well in order to determine negative populations in FACS plots. All gates were based on this method. We mentioned it in the methods section (**page 23, lines 6 and 7**).

8. Suggestion to examine BCL6 expression levels in WT and *Cic* null CD4⁺PD-1^{hi}CXCR5⁺ T_{FH} cells.

It would be more informative to look at Bcl6 expression within the CXCR5+PD1hi population.

We have already shown that *Bcl6* mRNA levels were comparable between WT and *Cic* null T_{FH} cells (**Fig. 6a**). To make this point clearer, we analyzed MFI of BCL6 expression in WT and *Cic* null CD4⁺PD-1^{hi}CXCR5⁺ T_{FH} cells. Consistent with the previous qRT-PCR data (**Fig. 6a**), the levels of BCL6 were comparable between WT and *Cic* null CD4⁺PD-1^{hi}CXCR5⁺ T_{FH} cells. We presented this data in **Fig. 2d**.

9. Question for expression profiles of IL-21 and IFN γ in *Cic* null T_{FH} cells.

Are CXCR5+PD1hi cells in *Cic* KO secreting IL-21 or IFN γ ?

As described in the answer for the question #2, we found that *Cic* null T_{FH} cells express IL-21 and IFN γ and that the frequency of IFN γ ⁺ T_{FH} cells was much higher in *Cic* deficient mice than in WT mice (**Fig. 2e**).

10. Concern on the statement that CIC is not a T helper subset-specifying factor.

Line 102: they comment that CIC is not a T helper subset-specifying factor but IFN γ and TNF α are increased as well as FoxP3+CD25- CD4 T cells, suggesting CIC is having an effect on Th1 cells and Tregs.

We agree with the reviewer's opinion. We amended the text accordingly (**page 7, line 2**).

11. Concern on the statement that the absence of CIC relieves the normal requirement for CD28 co-stimulation.

Line 146: the statement “the absence of CIC relieves the normal requirement for CD28 co-stimulation” is not correct as proliferation was still increased in *Cic* KO compared to WT T cells in absence of anti-CD28.

This statement might have caused confusion. We replaced this sentence with “a strong co-stimulatory signal through CD28 overrides the enhanced TCR response in *Cic* deficient T cells” (page 9, lines 11 and 12).

12. Question for additional signaling components required for ETV5-mediated induction of *c-Maf* expression in CD4⁺ T cells treated with IL-6 and IL-21.

Why was addition of IL-6 and IL-21 ie Tfh-inducing conditions required to see c-maf expression? Why is *Etv5* overexpression not enough?

As described in the answer for the reviewer #1’s question #8, we found that activation of JAK2-STAT3 pathway is required for induction of *c-Maf* expression by ETV5 in CD4⁺ T cells (Supplementary Fig. 13).

13. Question about why mRNA levels of other T_{FH}-related genes were not increased in *Cic* deficient T_{FH} cells.

Why were other markers of Tfh cells such as *Bcl6*, *Il21* and *Cxcr5* not increased on *Cic* KO Tfh cells (Figure 6a)?

In this qRT-PCR analysis for expression profiles of 14 different genes critical for T_{FH} cell differentiation and function (Fig. 6a), we used FACS-sorted fully differentiated PD-1^{hi}CXCR5^{hi} T_{FH} cells. It would be reasonable to think that expression profiles of most genes determining T_{FH} cell characteristics should be similar between mature WT and *Cic* null T_{FH} cells. Therefore, the finding that the levels of *Bcl6*, *Il21* and *Cxcr5* were comparable between WT and *Cic* null T_{FH} cells could be appreciated. Furthermore, this result suggests that expression of not only *c-Maf* but also *Batf*, *Icos* and *Ifng* might be specifically regulated by CIC in T_{FH} cells.

14. Suggestion to analyze expression profiles of cytokines and MFI of the transcription factors in *in vitro* differentiated WT and *Cic* deficient T helper cells.

Supplementary Figure 4: need to include a Th0 culture to determine the background in *Cic* KO cells. What about IL-4, IFN γ , IL-17 under these conditions? What happens when you culture under Tfh conditions (IL-6 and IL-21) and look at *Bcl6* and *Il21* expression in *CIC* KO compared to WT CD4 T cells? It may be more informative to include MFI of the expression of the transcription factors. How many times was this experiment performed? n = ?.

As suggested, we examined expression profiles of IFN γ , IL-4 and IL-17 in WT and *Cic* null CD4⁺ T cells incubated under Th0, Th1, Th2 and Th17 conditions (**Supplementary Fig. 7a**). We also analyzed MFI of the expression of the transcription factors including T-bet, GATA3 and ROR γ t in the *in vitro* differentiated T helper cells. The data are presented in **Supplementary Fig. 7b**. We performed more than three independent experiments.

We examined levels of *Bcl6* and *Il21* in WT and *Cic* null CD4⁺ T cells cultured under T_{FH}-like condition (in the presence of IL-6 and IL-21). As shown below, their expression levels were comparable between WT and *Cic* null T cells, consistent with the qRT-PCR result that the levels of *Bcl6* and *Il21* were comparable between WT and *Cic* null T_{FH} cells (**Fig. 6a**).

Reviewer #3:

We thank the reviewer for the positive comments and helpful suggestions.

Major questions:

1. Suggestion to investigate autoimmune phenotypes in T cell-specific *Cic* null mice.

According to the data, *Vav1*-cre mice demonstrated more profound immune activation than that of *Cd4*-cre mice. Unfortunately, it was not clearly whether *Cd4*-cre mice also developed tissue inflammation and systemic autoimmunity as *Vav1*-cre mice did. What is the exact mechanism for CIC to prevent autoimmunity? How about the innate immune system without CIC?

Accordingly, we examined whether T cell-specific *Cic* null (*Cic*^{f/f}*Cd4*-Cre) mice develop systemic autoimmunity with age. Serum levels of anti-dsDNA antibody were increased (**Fig. 4c**) and infiltration of immune cells into lung, liver and kidney tissues was apparent in about 14-month-old *Cic*^{f/f}*Cd4*-Cre mice (**Fig. 4d and e**), demonstrating that loss of CIC in T cells eventually leads to systemic autoimmunity in mice. We also analyzed myeloid lineage cells such as macrophages and dendritic cells (DCs) in *Cic*^{f/f}*Vav1*-Cre mice. The numbers of macrophages and DCs were increased in spleen of *Cic*^{f/f}*Vav1*-Cre mice at 10 weeks of age (**Fig. 1b**). However,

expression profiles of co-stimulatory molecules, such as CD80, CD86, CD40 and ICOSL, on the surface of DCs were comparable between WT and *Cic^{f/f}Vav1-Cre* mice (**Supplementary Fig. 4**).

2. Question about relationship between the enhanced T_{FH} cell differentiation and T cell hyperactivation in *Cic* deficient mice.

Is the aberrant Tfh differentiation primarily responsible for the Cd4 hyperactivation? This could be evaluated by crossing Cd4-cre mice with certain knockout mice, such as Icos KO, SAP KO or cMaf KO.

This is a good question. The suggested experiments are ideal to address this question. However, it will take a couple of years to generate and analyze those compound mutant mice. We hope to explore this interesting question in our follow-up study of this story. On the other hand, *in vitro* T cell activation assay has shown that *Cic* null CD4⁺ T cells more sensitively respond to TCR stimulation than WT cells (**Fig. 3c and d**), suggesting that hyperactivation of T cells in *Cic* deficient mice might be in part due to the hypersensitive TCR response.

3. Suggestion to examine thymic negative selection in *Cic* deficient mice.

The authors show CIC-deficiency affects TCR signaling. Will it affect the negative selection in the thymus? Defective negative selection could also lead to more Tfh differentiation in the peripheral but it becomes the second effect.

Accordingly, we assessed thymic negative selection in WT and *Cic^{f/f}Cd4-Cre* mice. The rates of TCR-induced apoptosis in double positive (DP) thymocytes and Nur77 expression in CD4⁺ thymocytes were comparable between WT and *Cic^{f/f}Cd4-Cre* mice, suggesting that the thymic negative selection might not be affected by *Cic* deficiency. The data are presented in **Supplementary Fig. 9**.

4. Question about functionality of *Cic* null Treg cells.

It was striking that majority of Treg cells were CD25- in the absence of CIC. Have the function of Treg cells been examined?

As described in the answer for the reviewer #1's question #2, we found that the suppressive activity and the expression profiles of co-stimulatory molecules are comparable between WT and *Cic* deficient CD4⁺ Treg cells (**Supplementary Fig. 8b, c and d**).

5. Suggestion to examine knock-down efficiency of shRNAs for *Etv5* and *c-Maf* in WT and *Cic* deficient CD4⁺ T cells.

There were no data to show the expression levels of overexpression or knockdown of ETV5 and cMaf. Without knowing the relevance of these manipulations as compared to the expression levels of WT and CIC-deficient T cells, it is difficult to judge whether such results are physiological.

We thank the reviewer for this meaningful suggestion. Accordingly, we examined levels of ETV5 and c-MAF in WT and *Cic* null CD4⁺ T cells infected with control, shETV5 or shMAF expressing retrovirus by western blot analysis. The image is presented in **Supplementary Fig. 11b**.

Minor questions:

The authors concluded increased B cells were responsible to splenomegaly and lymphadenopathy. How about the numbers of myeloid cells?

Accordingly, we measured the numbers of macrophages and DCs in spleen of WT and *Cic*^{f/f}*Vav1-Cre* mice at 10 weeks of age. The data are included in **Fig.1b**.

The authors concluded there was an increased Th1 differentiation. According to Fig 1C, almost a 2-fold increase of CD44⁺ effector/memory CD4 T cells but in Fig S3C, the increase of CD44⁺IFN γ ⁺ were much less than 2-fold. I would conclude actually there was less Th1 differentiation in total effector CD4 T cells in CIC-deficient mice.

Our conclusion on the increased proportion of CD44⁺IFN γ ⁺ T cells in *Cic* deficient mice was that loss of CIC elicits type-1 immune responses. On the other hand, the *in vitro* T helper cell differentiation assay has shown that loss of CIC did not affect efficiency of T_H1 cell differentiation (**Supplementary Fig. 7**).

References

1. Kim, H.-J., Verbinnen, B., Tang, X., Lu, L. & Cantor, H. Inhibition of follicular T-helper cells by CD8(+) regulatory T cells is essential for self tolerance. *Nature* **467**, 328–332 (2010).
2. Chang, J.-H. & Chung, Y. Regulatory T cells in B cell follicles. *Immune Netw.* **14**, 227–236 (2014).
3. Xie, Q. *et al.* Regulation of c-Maf and α A-Crystallin in Ocular Lens by Fibroblast Growth Factor Signaling. *J. Biol. Chem.* **291**, 3947–3958 (2016).
4. Betz, B. C. *et al.* Batf coordinates multiple aspects of B and T cell function required for normal antibody responses. *J. Exp. Med.* **207**, 933–942 (2010).
5. Ise, W. *et al.* The transcription factor BATF controls the global regulators of class-switch recombination in both B cells and T cells. *Nat. Immunol.* **12**, 536–543 (2011).
6. Stone, E. L. *et al.* ICOS coreceptor signaling inactivates the transcription factor FOXO1 to promote Tfh cell differentiation. *Immunity* **42**, 239–251 (2015).
7. Choi, Y. S. *et al.* ICOS receptor instructs T follicular helper cell versus effector cell differentiation via induction of the transcriptional repressor Bcl6. *Immunity* **34**, 932–946 (2011).
8. Lee, S. K. *et al.* Interferon- γ excess leads to pathogenic accumulation of follicular helper T cells and germinal centers. *Immunity* **37**, 880–892 (2012).
9. Bour-Jordan, H. & Bluestone, J. A. Regulating the regulators: costimulatory signals control the homeostasis and function of regulatory T cells. *Immunol. Rev.* **229**, 41–66 (2009).
10. Zelenay, S. *et al.* Foxp3+ CD25- CD4 T cells constitute a reservoir of committed regulatory cells that regain CD25 expression upon homeostatic expansion. *Proc. Natl. Acad. Sci. U. S. A.* **102**, 4091–4096 (2005).

Point-by-point response: comments made by reviewers are in **boxes** for ease of reference.

Reviewer #1:

We thank the reviewer for many constructive suggestions. We were able to come to more solid conclusion on the role of CIC in autoimmunity and T_{FH} cell differentiation by performing additional experiments.

1. Question for contribution of *Cic* deficiency in Treg cells to the autoimmune-related phenotypes in *Cic* mutant mice.

It is still unclear whether Tregs contribute to the phenotype observed in *CICf/f-Vav1Cre* or *CIC-CD4Cre* mice. The authors indicated that the increased GC responses *CIC* KO mice is not due to defect in Treg cells, since suppressive activity of *CIC* KO nTregs towards conventional CD4⁺CD25⁻ T cells is comparable to WT Tregs. However, the role of *CIC* in generation and function of inducible Tregs and Tfrs was not considered. It is possible that *CIC* regulates suppressive function of Tregs towards specific T helper subsets. On other hand, accumulation of Tfh cells even in the present of Tfrs indicates that *CIC* KO Tfh (or conventional CD4⁺ T cells) could be resistant to Treg suppression. The above possibilities have to be addressed in order to clarify whether phenotype in *CICf/f-CD4-Cre* mice is Tregs dependent or independent.

We thank the reviewer for raising this critical point. In the original manuscript, as the reviewer mentioned, we have already shown that *CIC* deficiency does not affect Treg suppressive activity toward conventional T cells and that expression profiles of several Treg cell-associated surface molecules are comparable between WT and *Cic* deficient Treg cells (**Supplementary Fig. 8**).

To directly determine whether *CIC* deficiency in Treg cells contributed to the systemic autoimmunity in *Cic* mutant mice, we generated and characterized Treg cell-specific *Cic* null mice (*Cic^{ff}Foxp3-YFP-Cre*). A specific deletion of *Cic* in FOXP3⁺ Treg cells did not result in any autoimmune-related phenotypes including T cell hyperactivation and increased frequency of T_{FH} and GC B cells (**Supplementary Fig. 12**), suggesting that *CIC* deficiency in Treg cells might not contribute to the onset of autoimmunity in *Cic* mutant mice. We also found that *CIC* deficiency induced FOXP3 expression in naïve peripheral CD4⁺ T cells and that the increased frequency of CD25⁻FOXP3⁺ T cells in *Cic^{ff}Cd4-Cre* mice was due to expansion of Helios⁻NRP1⁻ peripheral Treg (pTreg) cell population (**Supplementary Fig. 13**). Overall, our data suggest that the autoimmune-related phenotypes in *Cic^{ff}Vav1-Cre* or *Cic^{ff}Cd4-Cre* mice are mainly due to *CIC* deficiency in non-Treg cells.

2. Suggestion to analyze T_{FH} cell differentiation of adoptively transferred WT and *Cic* KO OT-II cells that are infected with either control or shMAF-expressing retrovirus at latter time points.

Regarding mechanism, it is still unclear how ETV5/c-Maf axis could control Tfh generation. Since c-Maf is required for IL-4 and IL-21 production it is possible that ETV5/c-Maf crosstalk contributes to Tfh cell maintenance and function rather to Tfh cell priming. This is probably why knockdown of c-Maf in CIC KO T cells (Fig. 6f) only moderately affected Tfh cell priming. Authors have to extend analysis to day 14 and 21 to define if it required for Tfh maintenance.

As the reviewer suggested, we carried out adoptive transfer experiments using WT and *Cic* null OT-II cells transduced with either control or shMAF-expressing retrovirus and analyzed T_{FH} cell differentiation at day 14 after immunization. In all three groups, the frequency of T_{FH} cells derived from the transferred OT-II cells was dramatically reduced, compared to that at day 7 (**Fig. R1a and Fig. 6f**). Moreover, differences in the proportions of T_{FH} cells among three groups disappeared at day 14 (**Fig. R1a**). These data indicate that T_{FH} cells derived from OT-II cells by ovalbumin immunization are not maintained for 14 days. We also found that the number of total OT-II cells transferred to recipient mice was decreased by about 60% at day 14 compared to that at day 7 (data not shown).

Consistent with our results, it has been reported that the number of transferred total OT-II cells and OT-II T_{FH} cells is highly increased at days 5-7 after immunization and decreased thereafter¹ (**Fig. R1b**). Moreover, Liu et al.² analyzed OT-II T_{FH} cells at days 2 and 6 as an early and fully differentiated stage, respectively. Taken together, the time point for our analysis of T_{FH} cell

Fig. R1. T_{FH} cells derived from OT-II cells are not maintained at day 14 after immunization.
 (a) WT Thy1.1⁺ OT-II cells infected with control retrovirus and *Cic* null Thy1.1⁺ OT-II cells infected with control or shRNA against *c-Maf* (shMAF) expressing retrovirus were transferred into Thy1.2⁺ B6 recipient mice. Fourteen days after immunization with NP-OVA in alum, the Thy1.1⁺ OT-II cells were analyzed for T_{FH} cell differentiation using flow cytometry. n=4 in each group.
 (b) Previously reported data showing that the numbers of transferred total OT-II and OT-II T_{FH} cells are dramatically reduced at day 14 after immunization. The image was taken from Figure 2a in Wang et al. 2014 *Nat. immunol.*

differentiation of OT-II cells in Fig. 6f is suitable for investigating the effect of *c-Maf* knock-down on T_{FH} cell maintenance.

3. Question on the reason why treatment with IL-6 and IL-21 did not induce expression of *c-Maf* in CD4⁺ T cells infected with either control or ETV5-overexpressing retrovirus.

Data in Fig. 6 E are not clear and contradict previous reports. Based on the data neither IL-6/IL-21 treatment nor ETV5 over-expression alone could induce *c-Maf* expression and presence of both components is required for *c-Maf* induction. However, it has been reported before that IL-6/IL-21 are potent *c-Maf* inducers. It is unclear why authors have different result.

We thank the reviewer for this question. As the reviewer pointed out, *c-Maf* expression was not significantly induced by treatment with IL-6 and IL-21 under our experimental conditions (**Fig. R2a**). In this experiment, naïve CD4⁺ T cells were purified and activated with anti-CD3/CD28 in the presence or absence of IL-6 and IL-21. Meanwhile, we infected the cells with control or ETV5-expressing retrovirus at 24h and 36h after TCR stimulation. Twelve hours after the second viral infection, cells were rested for 24h by addition of IL-7, and then re-stimulated with anti-CD3 for 2h.

We think that the reason for the treatment with IL-6 and IL-21 not substantially inducing expression of *c-Maf* is because the cells were too much activated under our experimental conditions. Consistent with this idea, we found that *c-Maf* expression was dramatically induced in CD4⁺ T cells cultured under T_{FH}-like condition (in the presence of IL-6 and IL-21) at 28h after TCR stimulation, but this induction gradually disappeared at later time points (**Fig. R2b**).

Fig. R2. Effect of duration of TCR stimulation on IL-6 and IL-21-mediated induction of *c-Maf* expression in CD4⁺ T cells.

(a) Relative levels of *c-Maf* mRNA in CD4⁺ T cells cultured in the presence (T_{FH}-like) or absence (neutral) of IL-6 and IL-21. qRT-PCR results for levels of *c-Maf* mRNA in control virus-infected CD4⁺ T cells (in Figures 6b and 6d) were re-analyzed.

(b) Relative levels of *c-Maf* mRNA in CD4⁺ T cells cultured under T_{FH}-like and neutral conditions at different time points. Error bars indicate SEM. *p < 0.05.

4. Suggestion to determine whether STAT3 is required for ETV5 binding to *c-Maf* promoter.

Authors claimed that STAT3 is required for ETV5 mediated expression of c-Maf. It will be interesting to further define the nature of this cooperation and determine whether STAT3 is required for ETV5 binding to c-Maf promoter.

We appreciate this excellent suggestion. Accordingly, we carried out ChIP in CD4⁺ T cells treated with either DMSO or Stattic (the STAT3 inhibitor) using anti-ETV5 antibody, and analyzed the enrichment of ETV5 binding to *c-Maf* promoter by qPCR. Inhibition of STAT3 activity did not affect ETV5 promoter occupancy of *c-Maf* gene, suggesting that ETV5 can bind to *c-Maf* promoter independent of STAT3 activity. We presented this ChIP-qPCR result in **Supplementary Fig. 16c**.

5. Suggestion to identify additional ETV5 target genes critical for regulation of T_{FH} cell differentiation.

Knockdown of ETV5 in CIC KO T cells (Fig. 6f) have significant impact on Tfh cell priming, suggesting that ETV5 has additional and distinct targets from c-Maf facilitating Tfh cell priming. This target(s) have to be defined to confirm that ETV5 drives Tfh differentiation in CIC KO mice.

Our study has demonstrated that *c-Maf* is a critical downstream target gene of CIC-ETV5 axis in regulation of T_{FH} cell differentiation by showing that knock-down of *c-Maf* expression significantly suppressed T_{FH} cell differentiation in *Cic* KO OT-II cells (**Fig. 6f**). Nevertheless, we agree that there could be additional ETV5 target genes that contribute to the enhanced T_{FH} cell differentiation in *Cic* deficient mice. We have already mentioned about this possibility in Discussion section. We believe that identification of other ETV5 target genes important for promotion of T_{FH} cell differentiation is beyond the scope of the current paper and would like to address this as a separate paper in the future by analyzing ETV5-mediated gene expression regulatory networks in the process of T_{FH} cell development and maintenance.

6. Suggestion to examine expression profiles of T helper cytokines in T_{FH} and non-T_{FH} cells in *Cic^{ff}Cd4-Cre* mice.

It will be interesting to examine expression of T helper cytokines in Tfh and non Tfh cells in CICf/f-CD4Cre mice. Authors for some reasons more extensively analyzed CICf/f-Vav1Cre mice compared to CICf/f-CD4Cre mice. Based on chemokine expression, CIC has intrinsic function in controlling Th1 and Th2 responses. In this regard, it is not clear why CIC does not impact the Th1 and Th2 development in vitro.

We appreciate this reviewer's comment. Accordingly, we examined expression profiles of 9 different cytokines in CD44⁺CD4⁺ T cells from WT and *Cic^{ff}Cd4-Cre* mice (**Supplementary Fig. 10c**), as we did in *Cic^{ff}Vav1-Cre* mice. We also examined the proportion of IL-21⁺, IL-4⁺

and IFN γ ⁺ T_{FH} cells in *Cic^{ff}Cd4-Cre* mice (**Supplementary Fig. 11a**). We found that the frequency of T_{H1} type or T_{H2} type cytokine-expressing CD4⁺ T cells was increased in *Cic^{ff}Cd4-Cre* mice. As the reviewer mentioned, CIC deficiency did not promote T_{H1} and T_{H2} cell differentiation *in vitro* (**Supplementary Fig. 7**). This discrepancy might be due to the difference between *in vitro* and *in vivo* T cell differentiation conditions; the *in vitro* T_H subset polarizing conditions might override the effect of CIC deficiency on T_H subset-specific differentiation, whereas complex *in vivo* environment might allow *Cic* null CD4⁺ T cells to preferentially develop into T_{H1} and T_{H2} cells as well as T_{FH} cells.

Reviewer #2:

We are grateful to the reviewer for the critical suggestions. We feel that the impact of our manuscript has been significantly raised.

1. Question for contribution of CD4⁺FOXP3⁺CD25⁻ T cells in autoimmune-related phenotypes in *Cic* mutant mice.

The majority of Tregs from *Cic*-deficient mice are CD4⁺Foxp3⁺CD25⁻, but they have yet to prove these cells in *Cic*-deficient mice are functional and can suppress and are thus not contributing to T cell activation and autoimmunity in *Cic*-deficient mice. Only CD4⁺CD25⁺ and not Foxp3⁺CD25⁻ CD4⁺ T cells were investigated in the elegantly laid out suppression assays in suppl Fig 8. This is an important point as a mechanism of suppression by Foxp3⁺CD25⁺ Tregs cells is consumption of IL-2, but this would not occur in the absence of CD25 expression ie in Foxp3⁺CD25⁻ Tregs and these form the majority of Treg compartment in *Cic*-deficient mice. Thus the question of whether CD4⁺Foxp3⁺CD25⁻ cells contribute to disease in *Cic*-deficient mice remains unanswered.

We thank the reviewer for raising this critical point. Although we did not compare Treg suppressive activity between WT and *Cic* deficient CD25⁻FOXP3⁺ T cells, we inferred from the data for expression profiles of Treg surface molecules on CD25⁻FOXP3⁺ T cells (Supplementary Fig. 8b) that WT and *Cic* deficient CD25⁻FOXP3⁺ T cells might have comparable functionality. To fundamentally address the reviewer's questions, we examined whether CIC deficiency in FOXP3⁺ Treg cells contributed to the autoimmune-related phenotypes in *Cic* mutant mice. To this end, we generated and characterized Treg cell-specific *Cic* null mice (*Cic^{ff}Foxp3-YFP-Cre*). These mice did not exhibit any autoimmune-related phenotypes (**Supplementary Fig. 12**), suggesting that the autoimmune-related phenotypes in *Cic* deficient mice are mainly due to CIC deficiency in non-Treg cells.

To gain better insight on the role of CIC in generation of CD4⁺CD25⁻FOXP3⁺ T cells, we further examined characteristics of CD25⁻FOXP3⁺ T cells in *Cic^{ff}Cd4-Cre* mice. We found that Helios⁻NRP1⁻CD25⁻FOXP3⁺ T cell population was dramatically increased in *Cic^{ff}Cd4-Cre* mice (**Supplementary Fig. 13c**), indicating that the increased frequency of CD25⁻FOXP3⁺ T cells in

Cic^{ff}Cd4-Cre mice was due to expansion of Helios⁺NRP1⁻ peripheral Treg (pTreg) cell population. Interestingly, we also found that CIC deficiency induced FOXP3 expression in naïve peripheral CD4⁺ T cells (**Supplementary Fig. 13a**) and that *Cic* deficient CD4⁺CD25⁻FOXP3⁺ T cells were composed of both CD44^{lo}CD62L^{hi} ‘naïve-like’ and CD44^{hi}CD62L^{lo} ‘effector-like’ populations, whereas WT cells were mainly CD44^{hi}CD62L^{lo} ‘effector-like’ cells (**Supplementary Fig. 13b**). Taken together, we concluded that CIC deficiency induces FOXP3 expression in naïve CD4⁺ T cells, thereby increasing the population of CD4⁺CD25⁻FOXP3⁺ T cells in the periphery.

2. Concern on the title and the focus of the manuscript.

The argument that CIC is specifically affecting Tfh cells and not other T helper cell subsets is misleading. Loss of CIC results in general T cell activation with an increase in effector and memory CD4⁺ T cells. While in vitro differentiation experiments of naïve cells show no difference in polarization to Th1, Th2 and Th17 cells, this was not shown for Tfh cell differentiation conditions. Furthermore, it is well known that other T helper cell subsets such as Th1, Th2, Th17 cells can also contribute to autoimmunity and/or B cell responses. Hence the title and the focus of the manuscript suggesting that Capicua specifically restrains Tfh cell differentiation is somewhat unfounded.

We appreciate this reviewer’s comment. We agree with the reviewer that CIC does not specifically regulate T_{FH} cell development. Considering the reviewer’s concern, we changed the title as “Capicua deficiency causes autoimmunity and promotes follicular T helper cell differentiation due to ETV5 de-repression”.

3. Question about functional differences in driving B cell responses between *Cic*-deficient and WT T_{FH} cells.

The authors performed additional experiments to find comparable expression of IL-21 and IL-4 between *Cic*-deficient and WT Tfh cells, but significantly more IFN γ expression by *Cic*-deficient Tfh cells. However, the question of functional differences in driving B cell responses between *Cic*-deficient and WT Tfh cells still remain unanswered.

We thank the reviewer for raising this point. To examine functional differences in driving B cell responses between WT and *Cic* null T_{FH} cells, we first examined expression profiles of IL-4, IL-21 and IFN γ in T_{FH} cells from WT and *Cic^{ff}Cd4-Cre* mice. As observed in *Cic^{ff}Vav1-Cre* mice (**Fig. 2e**), the frequency of IFN γ ⁺ T_{FH} cells was significantly increased in spleen of *Cic^{ff}Cd4-Cre* mice (**Supplementary Fig. 11a**). Next, we analyzed IgG2a⁺ GC B cells in WT and *Cic^{ff}Cd4-Cre* mice, because IFN γ is a cytokine that drives IgG2a class-switching in B cells during GC reactions³⁻⁵. The proportion and the number of IgG2a⁺ GC B cells were significantly increased in *Cic^{ff}Cd4-Cre* mice (**Supplementary Fig. 11b**). These findings indicate that CIC deficiency in T_{FH} cells promotes IgG2a class-switching in GC B cells due to the enhanced expression and

secretion of IFN γ . These data also explain why the serum IgG2a levels were increased in the *Cic* mutant mice (**Fig. 1d and 4b**).

References

1. Wang, H. *et al.* The transcription factor Foxp1 is a critical negative regulator of the differentiation of follicular helper T cells. *Nat. Immunol.* **15**, 667–675 (2014).
2. Liu, X. *et al.* Transcription factor achaete-scute homologue 2 initiates follicular T-helper-cell development. *Nature* **507**, 513–518 (2014).
3. Snapper, C. M. & Paul, W. E. Interferon-gamma and B cell stimulatory factor-1 reciprocally regulate Ig isotype production. *Science* **236**, 944–947 (1987).
4. Huang, S. *et al.* Immune response in mice that lack the interferon-gamma receptor. *Science* **259**, 1742–1745 (1993).
5. Lee, S. K. *et al.* Interferon- γ excess leads to pathogenic accumulation of follicular helper T cells and germinal centers. *Immunity* **37**, 880–892 (2012).

Point-by-point response: comments made by reviewers are in **boxes** for ease of reference.

Reviewer #1:

The revised manuscript is significantly improved and the authors have addressed almost all the concerns. However, it is likely that ETV5 has other targets in addition to c-Maf. Firstly, c-maf knockdown in CIC KO T cells partially reduced Tfh cell development. In this regard, we recommend the authors to clarify that c-maf is one of the potential targets of ETV5. Secondly, authors detected enhanced level of IFN γ in CIC KO Tfh cells compared to WT Tfh cells whereas IL-21 and IL-4 expression was intact. C-Maf is required for IL-4 and IL-21 production but not IFN γ . Thus, CIC or ETV5 may be additional targets responsible for IFN γ expression. This point has to be discussed in the paper.

We thank the reviewer for this suggestion. Accordingly, we discussed this point in the discussion section (page 18, first paragraph).

Reviewer #2:

Park et al have attempted to address the remaining concerns that were raised from the revised version of this manuscript. I still feel the role of capicua is to restrain general CD4+ T cell activation and in its absence the autoimmunity present is due to overall CD4+ T cell activation, but no doubt Tfh cells are playing a role in addition to potential other CD4+ helper T cells such as Th1, Th2, and Th17 cells.

We appreciate this reviewer's comment. As we mentioned in the previous rebuttal, we agree with the reviewer that CIC does not specifically regulate T_{FH} cell development. We will keep this comment in mind, and pursue the role of CIC in T cell activation in the future.